# Accelerating Newton-Schulz Iteration for Orthogonalization via Chebyshev-type Polynomials

## Abstract

The problem of computing optimal orthogonal approximation to a given matrix has attracted growing interest in machine learning. Notable applications include the recent Muon optimizer or Riemannian optimization on the Stiefel manifold. Among existing approaches, the Newton-Schulz iteration has emerged as a particularly effective solution, as it relies solely on matrix multiplications and thus achieves high computational efficiency on GPU hardware. Despite its efficiency, the method has inherent limitations—its coefficients are fixed and thus not optimized for a given matrix. In this paper we address this issue by proposing a Chebyshev-optimized version of Newton-Schulz (CANS). Based on the Chebyshev's alternance theorem, we theoretically derive optimal coefficients for the 3-rd order Newton-Schulz iteration and apply a Remez algorithm to compute optimal higher-degree polynomials. We leverage these polynomials to construct controlled approximate orthogonalization schemes, which is of interest in deep learning applications. Practically, we demonstrate the method's effectiveness in two key applications: orthogonalization in the Muon optimizer, and providing an efficient retraction alternative for Riemannian optimization on the Stiefel manifold.

## 1 Introduction

Polar decomposition of a matrix $X \in \mathbb{R}^{m \times n}, m \geq n$ is a factorization $X = WH$, where $W \in \mathbb{R}^{m \times n}$ has orthonormal columns and $H \in \mathbb{R}^{n \times n}$ is a positive semidefinite symmetric matrix (or Hermitian in the complex case). An important application of the polar decomposition is the orthogonal Procrustes problem:

$$\min_{Q:\, Q^T Q = I} \|Q - X\|_F,$$

with the solution being $Q = W$ the polar factor of $X$. For generalization, see (Schönemann, 1966).

Polar decomposition can be computed directly using the singular value decomposition $X = USV^T$, which immediately leads to $W = UV^T, H = VSV^T$. However, calculating the SVD can be costly for many applications. There are several iterative methods available, including Newton (Kenney & Laub, 1992) and Halley's methods (Nakatsukasa et al., 2010), which require matrix inversion. In this work, we consider the Newton-Schulz iteration (Björck & Bowie, 1971; Kovarik, 1970; Higham, 2008), which only requires matrix multiplication:

$$X_{k+1} = \frac{3}{2}X_k - \frac{1}{2}X_k X_k^T X_k, \quad X_1 = X. \tag{1}$$

This iteration converges to the orthogonal factor of the polar decomposition if $\sigma_1(X) < \sqrt{3}$ and $\sigma_n(X) > 0$. Classical Newton-Schulz iteration can be also extended to higher degrees (Bernstein & Newhouse, 2024):

$$X_{k+1} = \alpha_1^k X_k + \alpha_3^k X_k X_k^T X_k + \alpha_5^k X_k (X_k^T X_k)^2 + \cdots + \alpha_{2t+1}^k X_k (X_k^T X_k)^t,$$

which can be rewritten using SVD of $X_k = U S_k V^T$ as follows:

$$X_{k+1} = U(\alpha_1^k S_k + \alpha_3^k S_k^3 + \alpha_5^k S_k^5 + \cdots + \alpha_{2d+1}^k S_k^{2d+1})V^T = U p_k(S_k) V^T.$$

In order for these iterations to converge to the orthogonal polar factor, the composition of polynomials $p_k(p_{k-1}(\ldots p_1(x)))$ should converge to the unity function $f \equiv 1$ on the segment $[\sigma_n(X), \sigma_1(X)]$. Indeed, the desired property is:

$$
\begin{aligned}
\|X_{k+1} - UV^T\|_2 = \|U(p_k(S_k) - I)V^T\|_2 &= \|p_k(S_k) - I\|_2 \\
&= \|p_k(p_{k-1}(\ldots p_1(S))) - I\|_2 \\
&= \max_i |p_k(p_{k-1}(\ldots p_1(s_i))) - 1| \\
&\leq \max_{s \in [\sigma_n, \sigma_1]} |p_k(p_{k-1}(\ldots p_1(s))) - 1| \to 0, \quad k \to \infty,
\end{aligned}
\tag{2}
$$

where we used orthogonal invariance of the spectral norm. However, in some applications (e.g., Muon optimizer), high orthogonalization accuracy may not be necessary and finding an approximation of $f \equiv 1$ with an error $\varepsilon$ is sufficient. This allows to balance between accuracy and efficiency when selecting polynomials.

In this work, we propose algorithms for optimizing the coefficients of the classical Newton-Schulz method, based on the Chebyshev alternation theorem. This framework, which we call *Chebyshev-accelerated Newton-Schulz (CANS)*, enables us to obtain polynomials with the desired properties and accelerated convergence. Our main contributions are:

- We derive theory for finding odd polynomials that optimally approximate the unity function on a given segment $[a, b]$ (Section 3.1). This leads us to explicit formulas when $p_k$ are of degree 3 and Remez algorithm for larger degrees. Given the bounds on the singular values, these polynomials lead to methods that outperform Newton-Schulz (Section 3.2).

- We develop new polynomials that are confined within the interval $[1 - \delta, 1 + \delta]$ with a user-specified $\delta$ (inexact orthogonalization), while maximizing the polynomial derivative in the vicinity of zero (Section 4). This is motivated by the needs of the orthogonalization procedure of the *Muon* optimizer (Jordan et al., 2024b). For the same target $\delta$, our polynomials achieve a larger derivative compared to original Muon polynomial and those from (Jiacheng, 2024), and yield faster convergence of the optimizer when training the NanoGPT (Section 5.2).

- We further demonstrate that by maximizing the derivative at the origin, our inexact orthogonalization polynomials can serve as an effective preprocessing step for an iterative method of choice. This is particularly useful when information about the smallest singular value is not available. We also show that the largest singular value can be accurately approximated via Gelfand's formula with negligible computational overhead (Section 3.3).

- In Section 5.3, we demonstrate the application of CANS for building an efficient retraction on the Stiefel manifold, which speeds up training of WideResNet with orthogonal constraints.

## 2 RELATED WORK

**Iterative methods.** First iterative method for the orthogonal polar factor, based on Taylor series expansion, was introduced in (Björck & Bowie, 1971; Kovarik, 1970). The work (Higham & Schreiber, 1990) developed an algorithm balancing inversion and multiplication. Subsequent methods like scaled Newton (Higham, 2008), Halley's method, QDWH (Nakatsukasa et al., 2010), and Zolo-pd (Nakatsukasa & Freund, 2016) improved convergence but require matrix inversion or QR, which is less GPU-friendly than pure matrix multiplications. The stability of these methods is analyzed in (Nakatsukasa & Higham, 2012). Scaling of Newton-Schulz iteration was explored in (Chen & Chow, 2014b;a). Notably, the polynomials derived in (Chen & Chow, 2014b) align with our formula for optimal third-degree polynomials, although our approach is applicable for higher degree polynomials. Concurrently with our work, Amsel et al. (2025) also studied optimal polynomials for the Newton-Schulz iteration. They independently derived the same optimal third-order polynomial (which also matches the formula in (Chen & Chow, 2014a)) and the same recursive scheme for polynomial composition (see Eq. 4). While Amsel et al. (2025) prove the optimality of such composition, their method and analysis is restricted to the exact case. In contrast, our work focuses primarily on the inexact case, introducing a method to construct polynomials that satisfy a given tolerance $\delta$ while also maximizing derivatives at zero to accelerate the convergence of smaller singular values. A further distinction concerns the use of Gelfand's formula.

**Deep learning.** In neural networks, Newton-Schulz iteration is applied for enforcing orthonormality of the weight matrices (Anil et al., 2019). Its computational efficiency has made it particularly valuable for optimizers requiring orthogonalization, including Muon (Jordan et al., 2024b; Bernstein & Newhouse, 2024) and Scion (Pethick et al., 2025). Related approaches have employed Newton iteration for computing matrix p-th roots in other optimizers (Anil et al., 2020).

**Riemannian optimization.** In Riemannian optimization on the Stiefel manifold, polar decomposition is one of the possible retractions (Absil et al., 2009) to the manifold, alongside Cayley transform (Li et al., 2020; Zhu, 2017; Gao et al., 2021) and QR.

## 3 OPTIMAL ODD POLYNOMIALS AND NEWTON-SCHULZ ITERATIONS

### 3.1 OPTIMAL ODD POLYNOMIALS

As stated in equation 2, our goal is to find an odd polynomial that best approximates the unity function $f \equiv 1$ on a given segment, in which the singular values of the matrix fall $[\sigma_n(X), \sigma_1(X)] \in [a, b]$.

By $L_n$ we shall denote the space of odd polynomials of degree $2n - 1$, that is,

$$L_n = \{\alpha_1 x + \alpha_3 x^3 + \cdots + \alpha_{2n-1} x^{2n-1} : \alpha_1, \alpha_3, \ldots, \alpha_{2n-1} \in \mathbb{R}\}.$$

Note that $\dim L_n = n$. Now fix $0 < a < b$ and $n \in \mathbb{N}$. We endow the space $C[a, b]$ with its standard norm, i.e. $\|f\|_{C[a,b]} = \max_{t \in [a,b]} |f(t)|$. For a function $f \in C[a, b]$ we consider the problem of finding $p \in L_n$ such that $\|f - p\|_{C[a,b]} = \min\{\|f - q\|_{C[a,b]} : q \in L_n\}$. A polynomial $p$ with the foregoing property we shall call *the best uniform odd polynomial approximation* of $f$ of degree $2n-1$. Since we do not consider approximations in any other sense, we shall use a shorter term *best odd polynomial approximation* omitting the explicit mention of the degree, if it is clear from the context. The powerful method of studying best polynomial approximations is provided by the Chebyshev equioscillation theorem (see (Trefethen, 2020, Section 10) for classical formulation, and (Hörmander, 2018, Theorem 5) for the general version). In our case it reduces to the following fact.

**Theorem 1.** *Let $0 < a < b$, $n \in \mathbb{N}$, and $f \in C[a, b]$. Then the following statements hold.*

*(i) The best odd polynomial approximation of $f$ is unique.*

*(ii) $p \in L_n$ is the best odd polynomial approximation of $f$ of degree $2n - 1$ if and only if there exist points $x_0 < x_1 < \cdots < x_n$ on the interval $[a, b]$ such that $|p(x_j) - f(x_j)| = \|p - f\|_{C[a,b]}$ for all $j = 0, \ldots, n$ and $p(x_j) - f(x_j) = -(p(x_{j-1}) - f(x_{j-1}))$ for all $j = 1, \ldots, n$.*

*Proof.* See Appendix A. □

The points $x_0, \ldots, x_n$ from Theorem 1 are said to form the *Chebyshev alternance for $p - f$*.

We shall need further properties of the best odd polynomial approximation of the unity function $f \equiv 1$. Given $0 < a < b$ and $n \in \mathbb{N}$ we denote by $p_{n,a,b}$ the best degree $2n - 1$ odd polynomial approximation of the unity on the interval $[a, b]$ and by $\varepsilon(n, a, b)$ we denote the value $\|p_{n,a,b} - 1\|_{C[a,b]}$. The following proposition contains basic properties of $p_{n,a,b}$.

**Proposition 1.** *Let $0 < a < b$ and let $n \in \mathbb{N}$. Then the following statements hold.*

*(i) If $x_0 < \cdots < x_n$ are the points of the Chebyshev alternance for $p_{n,a,b} - 1$, then $x_0 = a$ and $x_n = b$.*

*(ii) If $\varepsilon = \|p_{n,a,b} - 1\|_{C[a,b]}$, then $p_{n,a,b}(x_j) = 1 - (-1)^j \varepsilon$ for all $j = 0, \ldots, n$.*

*(iii) The derivative $p'_{n,a,b}(x)$ attains a local maximum at $x = 0$ and decreases on the interval $[0, x_1]$. Moreover, $p'_{n,a,b}(0) \geq (1 - \varepsilon)/a$.*

*(iv) For any $t > 0$ we have $\varepsilon(n, ta, tb) = \varepsilon(n, a, b)$ and $p_{n,ta,tb}(tx) = p_{n,a,b}(x)$.*

*Proof.* See Appendix B. □

Using the foregoing proposition it is easy to find a closed-form expression for $p_{2,a,b}$.

**Proposition 2.** *Let* $0 < a < b$. *Then*

$$p_{2,a,b} = \frac{2}{2\left(\frac{a^2+ab+b^2}{3}\right)^{3/2} + a^2b + b^2a}\left(\left(a^2+ab+b^2\right)x - x^3\right).$$

*Moreover, this polynomial attains its maximum on* $[a,b]$ *at* $x = e = \sqrt{\left(a^2+ab+b^2\right)/3}$. *Finally,*

$$\varepsilon(2,a,b) = \|p_{2,a,b} - 1\|_{C[a,b]} = \frac{2\left(\frac{a^2+ab+b^2}{3}\right)^{3/2} - a^2b - b^2a}{2\left(\frac{a^2+ab+b^2}{3}\right)^{3/2} + a^2b + b^2a}. \tag{3}$$

*Proof.* see Appendix C. $\square$

For the polynomials of higher degree, finding explicit formulas seems to be unrealistic, as the problem reduces to finding roots of polynomials of degree more than 4. Also we were not able to construct any transcendental formula for $p_{n,a,b}$. However, we can use an adaptation of the well-known Remez algorithm (see, e.g. (Trefethen, 2020, Section 10)) for finding optimal polynomials of higher degrees. We describe Remez algorithm in Appendix F.

### 3.2 Newton-Schulz iterations based on optimal odd polynomials

We outline several reasonable choices of polynomials for Newton-Schulz iterations of a matrix $X$.

At first we consider the case when we are given a priori estimates on the singular values of $X$, i.e. $a \le \sigma_k(X) \le b$ for all $k = 1, \ldots, n$. In this case it is natural to consider an integer $d_0 \in \mathbb{N}$ and an optimal odd polynomial $p_{d_0,a,b} = \sum_{k=1}^{d_0} \alpha_{2k-1} x^{2k-1}$. All singular values of the matrix $X_1 = \sum_{k=1}^{d_0} \alpha_{2k-1} X(X^T X)^{k-1}$ are contained in the interval $[a_1, b_1] = [1 - \varepsilon(d_0, a, b), 1 + \varepsilon(d_0, a, b)]$. Thus, we can again choose an integer $d_1$ (possibly distinct from $d_0$) and repeat this process with $p_{d_1,a_1,b_1}$ and matrix $X_1$. If $d_0, d_1, \ldots$ are chosen to be greater or equal than 2, then this process converges to the orthogonal factor $UV^T$ of $X$ in its polar decomposition (Algorithm 1). We present analysis of the convergence of these iterations in case of polynomials of degree 3 ($d_i = 2$).

Let $0 < a < b$ and consider the following recursion:

$$a_0 = a, \quad b_0 = b, \quad 0 < a < b$$
$$a_{n+1} = 1 - \varepsilon(2, a_n, b_n), \quad b_{n+1} = 1 + \varepsilon(2, a_n, b_n). \tag{4}$$

We also have $\varepsilon(d_k, a, b) = \|X_k - UV^T\|_2$.

**Proposition 3.** *With the definition above, the error of approximation* $\varepsilon_{n+1} = \varepsilon(2, a_n, b_n)$ *converges to zero quadratically. More precisely,*

$$\varepsilon_{n+1} \le \varepsilon_n^2 \quad and \quad \lim_{n\to\infty} \frac{\varepsilon_{n+1}}{\varepsilon_n^2} = \frac{3}{4}.$$

*Proof.* See Appendix D. $\square$

**Corollary 1.** *For the starting segment* $[a_0, b_0]$, *where* $0 < a_0 < 1$ *and* $b_0 = 1$, *the number of iterations necessary to achieve the desired error of approximation* $\varepsilon$ *in the spectral norm is as follows:*

$$n \ge \left\lceil \log_2\left(\frac{\ln \varepsilon}{\ln(1 - a_0)}\right) \right\rceil.$$

*Proof.* See Appendix E. $\square$

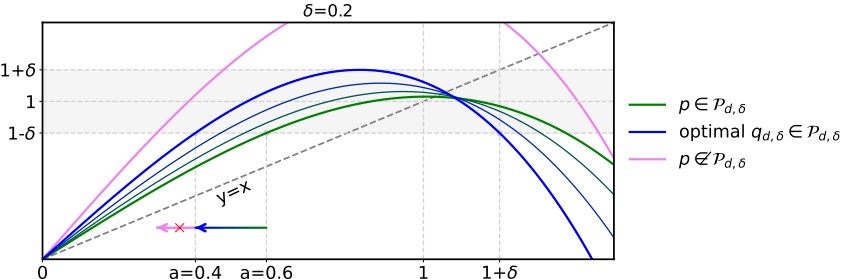

Figure 1: Illustration of the selection of a degree-3 ($d = 2$) polynomial with a large derivative at zero. The green polynomial falls into $[1 - \delta, 1 + \delta]$, but has insufficient derivative. The blue polynomial $q_{d,\delta}$ has the highest possible derivative among polynomials from $\mathcal{P}_{d,\delta}$. The purple polynomial is not part of $\mathcal{P}_{d,\delta}$, and its derivative is too large.

### 3.3 NORMALIZATION OF A MATRIX PRIOR TO NEWTON-SCHULZ ITERATIONS

To achieve the desired behavior of Newton-Schulz iterations (both classical Newton-Schulz and our modifications), one has to impose upper estimates for singular values of a matrix. That is, the first step of any algorithm based on Newton-Schulz is to normalize the matrix so that its singular values fall into the convergence range of polynomials (e.g. $(0, \sqrt{3})$ for classical NS, $[\varepsilon, 1]$ in our case). The easiest approach is to normalize by Frobenius norm, but this may significantly decrease small singular values and slow down the convergence. Ideally, the matrix should be normalized by its largest singular value. To estimate $\sigma_1$ efficiently, one may use power method (but it estimates $\sigma_1$ from below), randomized estimates (Dixon, 1983), (Halko et al., 2011, Lemma 4.1) or Gelfand's formula: $\sigma_1(A) \leq \|(A^T A)^k\|_F^{1/(2k)}$. If needed, the Gelfand's formula can be implemented without introducing extra matmuls because $(A^T A)^k$ is computed during Newton-Schulz iterations:

1. Compute matrices $(A^T A)^i$ for $i = 1...k$ and save them.

2. Compute $c = \|(A^T A)^k\|_F^{1/(2k)}$.

3. Compute $p_1(A/c) = (\sum_i^k \alpha_i (A^T A)^i / c^{2i})(A/c)$. Use $p_1(A/c)$ for the next iteration.

Note that for third-degree polynomials, we do not need to save extra matrices.

### 4 POLYNOMIALS WITH LARGE DERIVATIVES AT $x = 0$

Now we aim to construct polynomials that can rapidly uplift the smallest singular values, while deviating from 1 by no more than given $\delta$. It implies that they should have a large derivative at zero.

At first let us discuss the conditions that we impose on polynomials. Since it is desirable that the value $p(p(\ldots p(x) \ldots))$ falls into the interval $[1 - \delta, 1 + \delta]$ after sufficient number of iterations, it is natural to require that $p([1 - \delta, 1 + \delta]) \subset [1 - \delta, 1 + \delta]$. On the other hand, for $x \in [0, 1 - \delta]$ we want to guarantee, that $x$ is not moved further away from the desired interval. Hence, for $x \in [0, 1 - \delta]$ we require the condition $p(x) \geq x$. On the other hand, we do not impose any conditions on the behaviour of $p$ for $x > 1 + \delta$, thus we also need to add the restriction $p(x) \leq 1 + \delta$ for $x \in [0, 1 - \delta]$ (otherwise, we can not control the behaviour with respect to iterations of $p$). With the above considerations we introduce the set

$$\mathcal{P}_{d,\delta} = \{p \in L_d : x \leq p(x) \leq 1 + \delta \;\; \forall x \in [0, 1 - \delta], \;\; 1 - \delta \leq p(x) \leq 1 + \delta \;\; \forall x \in [1 - \delta, 1 + \delta]\}.$$

The problem posed at the beginning of the section can be now formulated as an optimization problem

$$\max_{p \in \mathcal{P}_{d,\delta}} p'(0). \tag{5}$$

We shall not solve this problem directly, but instead we replace it by another one, the solution of which can be reduced to the problem of finding best polynomial approximation of the unity function.

Consider for a polynomial $p \in \mathcal{P}_{d,\delta}$ the number $\mathfrak{a}_\delta(p) = \sup\{x \in [0, 1 - \delta] : p(x) < 1 - \delta\}$. That is, $\mathfrak{a}_\delta(p)$ is the left boundary of the biggest segment $[a, 1 + \delta]$ on which the values of a polynomial $p$ falls into $[1 - \delta, 1 + \delta]$. Intuitively, to increase the derivative of a polynomial $p$ at zero, we need to shift the left boundary $a$ of the described segment as close to zero as possible until there does not exist a polynomial that fits into the restrictions of $\mathcal{P}_{d,\delta}$ (see the shift from the green to the blue polynomial in Figure 1). Thus, we consider the optimization problem

$$\min_{p \in \mathcal{P}_{d,\delta}} \mathfrak{a}_\delta(p). \tag{6}$$

Below we show that the problem (6) has a unique solution that can be found explicitly for polynomials of degree 3 and by binary search for higher degrees (Algorithm 2). Moreover, we show the equivalence of problems (5) and (6) (i.e. optimal polynomials for these problems coincide) if $\delta$ is large enough.

**Proposition 4.** *Let $\delta \in (0, 1)$ and $d \in \mathbb{N}, d \geq 2$. Then the following statements hold.*

  *(i) There is a unique number $a = a(d, \delta) \in (0, 1 - \delta)$ such that $\varepsilon(d, a, 1 + \delta) = \delta$.*

  *(ii) The solution to the optimization problem (6) is unique, the minimum is equal to $a = a(d, \delta)$ from (i) and is attained on the polynomial $q_{d,\delta} = p_{d,a,1+\delta}$ (optimal odd polynomial on $[a, 1 + \delta]$ of degree $2d - 1$, see Section 3.1).*

  *(iii) The solution $q_{d,\delta}$ to the problem equation 6 satisfies the inequality $q_{d,\delta}(x) \geq cx$ for all $x \in [0, a(d, \delta)]$ with $c = (1 - \delta)/a(d, \delta) > 1$.*

  *(iv) Let $x_0 = a(d, \delta) < x_1 < \cdots < x_d = 1 + \delta$ denote the alternance points for the polynomial $q_{d,\delta}$. If $x_2 \geq 1 - \delta$, then $q_{d,\delta}$ is the solution to the problem in (5), i.e. it maximizes the derivative at zero on the set $\mathcal{P}_{d,\delta}$.*

*Proof.* See Appendix G. $\qquad\qquad\qquad\qquad\qquad\qquad\qquad\qquad\qquad\qquad\qquad\qquad\square$

Using a sequence of different polynomials, rather than iterating a single one, can push singular values into the target interval $[1 - \delta, 1 + \delta]$ more effectively and produce a faster-growing derivative at zero. The composition of polynomials can be constructed as follows:

  1. Start with the target $\delta \in (0, 1)$.
  2. Choose a degree $d_1 \in \mathbb{N}$ and find a larger interval $[1 - \delta_1, 1 + \delta_1]$ that a polynomial $p_1$ can map into $[1 - \delta_1, 1 + \delta_1]$ (in other words, $\varepsilon(d_1, 1 - \delta_1, 1 + \delta_1) = \delta$).
  3. Repeat this, choosing yet another $d_2 \in \mathbb{N}$ and polynomial $p_2$ to map an even larger interval $[1 - \delta_2, 1 + \delta_2]$ into the previous $[1 - \delta_1, 1 + \delta_1]$. Repeat this process $l$ times.

It is easy to see that the composition $f(x) = p_1(p_2(\ldots p_l(x) \ldots))$ maps the interval $[1 - \delta_l, 1 + \delta_l]$ into $[1 - \delta, 1 + \delta]$. Moreover, $f$ monotonically increases on $[0, 1 - \delta_l]$ and satisfies $f(x) > x$ for all $x \in [0, 1 - \delta_l]$. After rescaling the argument by multiplying with $(1 + \delta)/(1 + \delta_l)$ we obtain a function $g(x) = f(x(1 + \delta_l)/(1 + \delta))$ that has similar properties to iteration of $q_{d,\delta}$ but with a crucial advantage: its derivative at zero is higher. For example, if $d_i = d$, then $g'(0) \geq \left(q'_{d,\delta}(0)\right)^l$.

Polynomials with high derivatives at zero can be applied to matrices with rapidly decreasing singular values before orthogonalization (Algorithms 1, 2). This helps to speed up orthogonalization (see Figure 2). The number of iterations $\ell$ can be chosen either in advance, based on the desired budget of matmuls (the muon case), or until convergence to the desired accuracy $\varepsilon$ (the orthogonalization case).

**Algorithm 1** Orthogonalization with CANS.

**Input** Normalized matrix $X \in \mathbb{R}^{n \times p}, p \leq n$; $[a, b]$ where singular values of $X$ lie; number of iterations $\ell$; polynomials' degrees $2d_i - 1$.

**if** a is unknown **then**
    $X, a, b =$
    $= \delta\text{-orthogonalization}(X)$
**for** i **in** $0 \ldots \ell$ **do**
    **if** $d_i = 2$ **then**
        $p_i, \varepsilon$ are found using Proposition 2
    **else**
        $p_i, \varepsilon = \text{remez}(a, b, 2d_i - 1)$
    $a, b = 1 - \varepsilon, 1 + \varepsilon$
$X = p_s(p_{s-1}(\ldots p_1(p_0(X))))$
**Return** $X$

---

**Algorithm 2** $\delta$-orthogonalization.

**Input** Normalized $X \in \mathbb{R}^{n \times p}, p \leq n$; right boundary $B$; degrees $2d_i - 1, i = 0 \ldots \ell$; desired $\delta$; $eps = 1e\text{-}7$.
$A_l, A_r = 0, 1 - \delta$
**while** $|\delta - \varepsilon| > eps$ **do**
    $a, b = (A_l + A_r)/2, B$
    **for** i **in** $0 \ldots s$ **do**
        $p, \varepsilon = \text{remez}(a, b, 2d_i - 1)$
        $a, b = 1 - \varepsilon, 1 + \varepsilon$
    **if** $\varepsilon < \delta$ **then**:
        $A_r = (A_r + A_l)/2$
    **else**
        $A_l = (A_l + A_r)/2$
$X = p(X)$
**Return** $X, 1 - \delta, 1 + \delta$

---

## 5 APPLICATIONS

### 5.1 ORTHOGONALIZATION

Let us consider the problem of computing the orthogonal polar factor of a matrix $A$. We compare the performance of the classical Newton-Schulz (equation 1) to the CANS method (Algorithm 1). To find the composition of 3-rd order polynomials, we use explicit formulas from Proposition 2, for the 5-th order polynomials – the Remez algorithm. The Figure 2 shows the convergence of these algorithms for a matrix $A \in \mathbb{R}^{1000 \times 1000}$ with entries sampled from $\mathcal{N}(0, 1)$.

We conclude that the iterations with tuned coefficients converge noticeably faster than the classical Newton-Schulz (matmuls are proportional to time, see Table 3). CANS algorithm performs better when the boundaries of the spectrum are determined more accurately. Overestimating the smallest singular value results in faster convergence than underestimating it. $\delta$-orthogonalization helps to accelerate the convergence of the algorithm, even if the smallest singular value is not available.

### 5.2 MUON OPTIMIZER

Muon (Jordan et al., 2024b) is a recently developed optimizer for matrix-shaped parameters in neural networks, that has shown promising results in improving convergence and training speed (Liu et al., 2025).The key idea of Muon is the orthogonalization of the momentum $M_k$:

$$M_k = \beta M_{k-1} + (1 - \beta)G_k,$$

$$W_k = W_{k-1} - \eta Ortho(M_k),$$

where $G_k$ is the gradient on the step $k$, $M_k$ is the momentum, $W$ are the parameters that we wish to update, $\eta$ is the learning rate, $Ortho(M_k) = \arg\min_O\{\|M_k - O\|_F : O^T O = I \text{ or } OO^T = I\}$ (which is known as Procrustes problem with exact solution being polar factor $O = UV^T$ of $M_k = USV^T$). However, due to the prohibitive cost of SVD, authors instead choose to apply Newton-Schulz iteration with tuned coefficients for approximate orthogonalization. Authors empirically find, that in practice the singular values of the resulting matrix may deviate from 1 without harming the performance of optimizer for small models (for original Muon polynomial (Jordan et al., 2024b) the singular values fall into $[0.7, 1.2]$). However, further investigation suggested that decreased deviation improves the performance for larger models, e.g. GPT-2 medium (Cesista et al., 2025). In addition, higher derivative of composition of polynomials in zero $\phi(0)'$ noticeably improves the performance. Thus, the objective is to find composition $\phi(x)$:

$$\phi(x) = p_n(p_{n-1}(\ldots p_1(x))) \in [1 - \delta, 1 + \delta], \ s.t. \ \phi(0)' \to \max.$$

Prior works (Cesista et al., 2025; Jiacheng, 2024) have attempted to construct such polynomials using computational search. However, our theory allows to find optimal polynomials with these constraints.

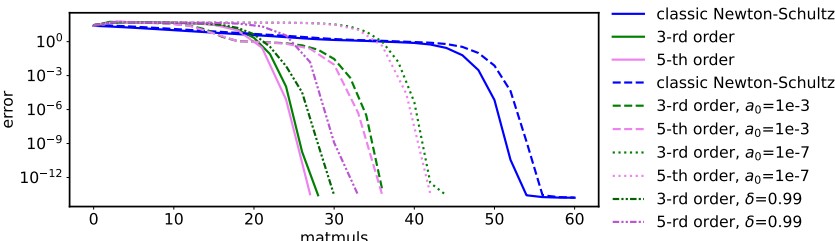

Figure 2: Convergence of iterative algorithms for matrix orthogonalization. The solid lines show the performance when the exact values of $\sigma_1(A), \sigma_n(A)$ are known, and the matrix is normalized by $\sigma_1(A)$. In other cases, the matrix is normalized by $\|(A^T A)^2\|_F^{1/4}$ and the precise value of the left boundary is $\sigma_n(A)/\|(A^T A)^2\|_F^{1/4}$=9e-5. The striped lines show performance for overestimated boundary $a_0$=1e-3, the dotted lines – for underestimated $a_0$=1e-7. The dashdotted lines show convergence of algorithm with 4 iterations of $\delta$-orthogonalization (Algorithm 2).

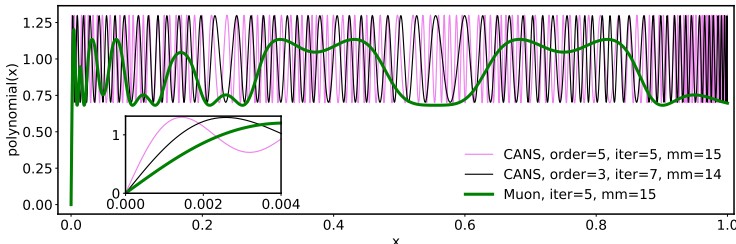

Figure 3: Comparison of CANS with the original Muon polynomial. Zoomed plot shows behavior near zero. "iter" denotes number of polynomials in composition, "mm" - total number of matmuls.

We have set the deviation $\delta = 0.3$ and generated a composition of 5 polynomials of 5-th degree (purple) and 7 polynomials of 3-rd degree (black), which are shown in Figure 3. Both polynomials have higher derivative at zero than original Muon polynomial $p(x) = 3.4445x - 4.7750x^3 + 2.0315x^5$, while requiring no more matmuls. Compositions of 9 3-rd order polynomials for $\delta = 0.00188$ (purple) and $\delta = 0.00443$ (blue) (Figure 4) also have higher derivatives than (Jiacheng, 2024) polynomial found by computational search. Polynomials' coefficients are presented in Appendix J.

The performance of Muon optimizer with proposed polynomials is tested on the task of training NanoGPT (Jordan et al., 2024a) (see Appendix H for details). The convergence of Muon with different polynomials is shown in the Figure 5. We observe, that CANS polynomial requiring 12 matmuls (purple) outperforms Muon polynomial with the same number of matmuls (4 iterations, cyan). The difference in convergence may be more pronounced when training larger models.

### 5.3 RIEMANNIAN OPTIMIZATION ON THE STIEFEL MANIFOLD

Let us introduce the following definitions, based on (Absil et al., 2009; Li et al., 2020).

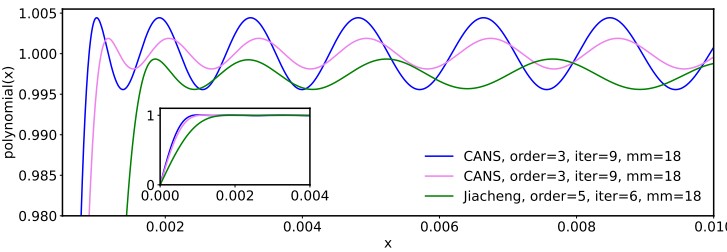

Figure 4: Comparison of CANS polynomials with (Jiacheng, 2024).

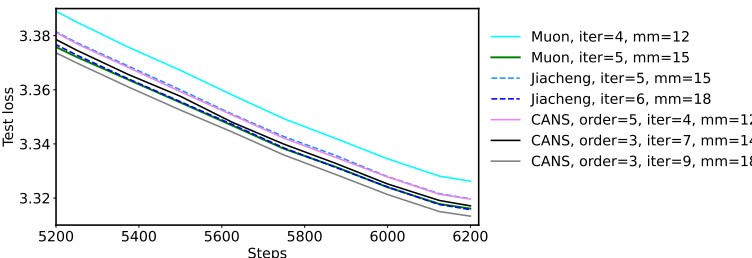

Figure 5: Test loss of NanoGPT trained using Muon optimizer with different polynomials.

**Definition 1.** *A Riemannian manifold $(\mathcal{M}, \rho)$ is a smooth manifold whose tangent spaces $T_x(\mathcal{M})$ are endowed with a smoothly varying inner product $\rho_x(\cdot, \cdot) : T_x(\mathcal{M}) \times T_x(\mathcal{M}) \to \mathbb{R}$, which is called the Riemannian metric.*

**Definition 2.** *A geodesic is a curve representing the locally shortest path between two points on manifold. An exponential map $Exp_x : T_x(\mathcal{M}) \to \mathcal{M}$ maps a tangent vector to the manifold. $Exp_x(tv)$ represents a geodesic $\gamma(t), t \in [0, 1]$, s.t. $\gamma(0) = x, \dot{\gamma}(0) = v$. A retraction is a smooth mapping from the tangent bundle to the manifold $Retr_x : T_x(\mathcal{M}) \to \mathcal{M}$ iff $Retr_x(0) = x$ and $DRetr_x(0) = id_{T_x(\mathcal{M})}$, where $D$ denotes derivative. Usually, retraction is a computationally efficient alternative to exponential mapping.*

The Stiefel manifold is a Riemannian manifold, consisting of $n \times p, n \geq p$ matrices with orthonormal columns $\mathcal{M} = St(n, p) = \{X \in \mathbb{R}^{n \times p} : X^T X = I\}$. The tangent space of $\mathcal{M}$ is defined as:

$$T_X(\mathcal{M}) = \{Z : Z^T X + X^T Z = 0\}.$$

The projection on $\mathcal{M}$ can be written as:

$$\pi_X(Z) = Z - \frac{1}{2}X(Z^T X + X^T Z) = WX, \tag{7}$$

$$W = \hat{W} - \hat{W}^T, \quad \hat{W} = ZX^T - \frac{1}{2}X(X^T Z X^T). \tag{8}$$

The process of Riemannian optimization of the function $f$ on the manifold $\mathcal{M}$ can be split into three steps. At first, the gradient $\nabla f$ in the Euclidean space is projected onto tangent space $T_{X_k}(\mathcal{M})$ to obtain $\nabla_{\mathcal{M}} f(X_k) = \pi_{X_k}(\nabla f)$. Secondly, momentum $M_k$ is transported to $T_X(\mathcal{M})$ and combined linearly with $\nabla_{\mathcal{M}} f(X_k)$ to get the updated momentum $M_{k+1}$. Finally, $X_{k+1}$ is computed as a step along the curve on the manifold with initial direction $M_{k+1}$. Parameters can be updated using the exponential map and parallel transport of momentum, but due to the computational complexity of these methods, retraction and vector transport are often used instead.

Let $\xi_X, \eta_X \in T_X(\mathcal{M})$ be tangent vectors. The vector transport of $\xi_X$ along retraction map $\text{Retr}_X(\eta_X)$ can be computed as $\tau_{\eta_X} \xi_X = \pi_{\text{Retr}_X(\eta_X)}(\xi_X)$. The projection is a linear mapping, so the first two steps can be combined $M_{k+1} = \alpha \pi_{X_k}(\nabla f(X_k)) + \beta \tau_{M_k}(M_k) = \pi_{X_k}(\alpha \nabla f(X_k) + \beta M_k)$.

There are several retractions of vector $\xi$ in point $X$, that can be used in practice (Absil et al., 2009). **QR decomposition:** $\text{Retr}_X(\xi) = qr(X + \xi)$, where $qr(A)$ is the $Q$ factor from QR decomposition. **Cayley transform:** $\text{Retr}_X(\xi) = (I - \frac{1}{2}W(\xi))^{-1}(I + \frac{1}{2}W(\xi))X$, with $W(Z)$ denoted in 8. (Li et al., 2020) approximates closed-form Cayley transform using iterative algorithm. **Polar decomposition:** $\text{Retr}_X(\xi) = UV^T = (X + \xi)(I + \xi^T \xi)^{-1/2}$, where $USV^T = X + \xi$ is SVD decomposition. Note that this retraction is known to be of the second order (Absil et al., 2009; Gawlik & Leok, 2018).

In this work, we propose to approximate the polar retraction using Newton-Schulz iteration with carefully chosen polynomials. The step of the Riemannian gradient descent can be written as

$$X_{k+1} = \text{Retr}_{X_k}(\alpha \pi_X(\xi)). \tag{9}$$

To find the interval for estimation of the polynomial's coefficients, we should estimate the condition number $\sigma_p(A)/\sigma_1(A)$ of the matrix $A = X_k + \alpha \pi_X(\xi)$. Let us compute the Gram matrix:

$$A^T A = (X + W(\xi)X)^T(X + W(\xi)X) = I + X^T W(\xi)^T W(\xi)X.$$

Therefore, $\sigma_p(A) = \sqrt{\sigma_p(A^T A)} \geq 1$. Since $A$ has size $n \times p, p \leq n$ and $p$ nonzero singular values, it follows that $\sigma_1(A) \leq \sqrt{\|A\|_F^2 - (p-1)} = c$, which yields a highly accurate estimate in this setting. Thus, we can normalize $A$ by $c$, set $[a, b] = [1/c, 1]$ and perform CANS orthogonalization.

### 5.4 EXPERIMENTS

Following the work (Li et al., 2020), we benchmark the performance of Riemannian optimization methods on the task of training CNN with orthogonal constraints. We train Wide ResNet (Zagoruyko & Komodakis, 2016) on classification of CIFAR-10. The convolutional kernels $K \in \mathbb{R}^{c_{out} \times c_{in} \times k \times h}$ are reshaped into $p \times n = c_{out} \times (c_{in} \cdot k \cdot h)$ matrices, which are restricted to Stiefel manifold. We optimize these parameters using Riemannian SGD with momentum and Riemannian ADAM, using vector transport and proposed polar retraction (see Appendix I, H). Other parameters are optimized with standard SGD or ADAM.

Tables 2, 1 show that our method has the lowest per epoch training time among other retractions, while achieving the same accuracy. It has a simple explanation. To form the matrix $W \in \mathbb{R}^{n \times n}$ for Cayley retraction as in (Li et al., 2020), 3 matmuls are needed (see 8) and multiplying by $W$ has asymptotics $\mathcal{O}(n^2 p)$. Cayley retraction can also be done using the Woodbury formula with asymptotics $\mathcal{O}(np^2)$, but more matmuls (see Appendix I). In contrast, forming $\pi_X(\xi)$ using formula 7 requires 2 matmuls; multiplications with $n \times p, p \leq n$ matrix $A$ in CANS have asymptotics $\mathcal{O}(np^2)$.

Table 1: Accuracy and training time for Wide ResNet-16-10 on CIFAR-10 using Adam.

| Retraction | Accuracy | Time per epoch (s) |
|---|---|---|
| - | 94.68 | **35.0** |
| Cayley (Li et al., 2020) | 95.77 | 71.2 |
| Cayley (Woodbury) | 95.69 | 70.9 |
| QR | 95.57 | 61.7 |
| CANS | **95.82** | 45.1 |

## 6 CONCLUSION

This work presented efficient algorithms for deriving the theoretically optimal coefficients for Newton-Schulz iteration. The practical effectiveness of CANS was demonstrated in accelerating the computation of the unitary polar factor, orthogonalization in the Muon optimizer, and fast retraction on the Stiefel manifold. We believe that our method can be useful for other applications as well, as it provides a general-purpose framework for finding optimized polynomials with desired accuracy.

## 7 REPRODUCIBILITY STATEMENT

The experimental details are described in Sections 5.2, 5.4 and Appendix H. The coefficients of polynomials are presented in Appendix J. Our code is planned to be made public after publication.

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

## A  PROOF OF THEOREM 1 FOR ODD POLYNOMIALS

In a nutshell, the result follows from a generalized version of the Chebyshev equioscillation theorem from (Hörmander, 2018). It applies to function spaces where an element is guaranteed to be zero if it vanishes at sufficiently many distinct points – a condition that holds in our case of odd polynomials.

The generalized version of the Chebyshev equioscillation theorem (Hörmander, 2018, Theorems 4-5) states the following. Consider a compact metric space $X$ and an $n$-dimensional vector space $L \subset C(X)$. Assume that any $f \in L$ that has n distinct zeroes in $X$ is identically equal to zero. Then the following statements hold.

1. For all $g \in C(X)$ there is a unique best approximation to $g$ in the space $L$, i.e. a function $G \in L$ such that $\|g - G\|_{C(X)} = \min_{f \in C(X)} \|g - f\|_{C(X)}$.

2. Moreover, $G \in L$ is the best approximation to $g$ if and only if there exists a set $E \subset X$ that consists exactly of $n + 1$ points such that $\|g - G\|_{C(X)} = \|g - G\|_{C(E)} = \min_{f \in C(X)} \|g - f\|_{C(E)}$.

Applying (1): At first we show that this theorem is applicable to $L = L_n$ and $X = [a, b]$, where $0 < a < b$ and $L_n$ is the space of odd polynomials of degree $\leq 2n - 1$. Indeed assume that $f \in L_n$ has $n$ distinct zeroes in $[a, b]$. Then it also has $n$ distinct zeroes in $[-b, -a]$, so f has at least $2n$ distinct zeroes. As $\deg f < 2n$ we conclude that $f = 0$. Thus, the generalized Chebyshev equioscillation theorem implies that the best odd polynomial approximation $G \in L_n$ to $g \in C[a, b]$ is unique and that there exists $E \subset [a, b]$ that consists of exactly $n + 1$ point such that $G$ is the best approximation to $g$ in the sense of the norm $\| \cdot \|_{C(E)}$.

Applying (2): Now let $E = \{x_0, ..., x_n\}$, where $a \leq x_0 < \cdots < x_n \leq b$. It remains to describe the best approximation to $g$ on the set $E$. We claim that if $G \in L_n$ and $\varepsilon$ satisfy $(-1)^j \varepsilon = G(x_j) - g(x_j)$ for all $j$, then $G$ is the best approximation of $g$ on $E$ with error $|\varepsilon|$. Indeed, if $F \in L_n$ approximates $g$ with error $\leq |\varepsilon|$ on $E$, then $F - G$ has at least $n$ zeroes on $[a, b]$ counting multiplicity (because the sign of the difference $F(x) - G(x)$ is alternating on the points $x_0, \ldots, x_n$). As above this implies that $F - G = 0$ since this is an odd polynomial with $n$ positive roots and $\deg(F - G) < 2n$. On the other hand, the conditions on $G$ and $\varepsilon$ above can be considered as a square linear system of equations (on coefficients of $G$ and $\varepsilon$). It is easy to verify that the matrix of this system of linear equations is nonsingular, so such $G$ and $\varepsilon$ exist. Thus, the best approximation $G \in L_n$ to $g$ on $E$ is unique and is characterized by the fact that $G - g$ equioscillates on $E$. Thus, Theorem 1 is proved.

## B  PROOF OF PROPOSITION 1

*Proof.* To simplify the notation we denote $p_{n,a,b}$ simply by $p$ throughout this proof.

(i) At first we note that the polynomial $p'$ is not identically zero and vanishes at the points $x_1, \ldots, x_{n-1}$, as these points are extrema of the function $p - 1$ and lie in the interior of the interval $[a, b]$. Clearly, $p'$ is even, so it also vanishes at $-x_1, \ldots, -x_{n-1}$. As $\deg p' \leq 2n - 2$, $p'$ cannot have any other roots, and, in particular, $p'(x_0) \neq 0$ and $p'(x_n) \neq 0$. Therefore, $x_0$ and $x_n$ belong to the boundary of $[a, b]$, so the statement (i) is proved.

(ii) In order to prove (ii) it suffices to verify that

$$p(a) = 1 - \varepsilon,$$

as the values $p(x_j)$ are uniquely determined by the value $p(x_0)$ due to Theorem 1 (ii). Assume the contrary, i.e. that

$$p(a) = 1 + \varepsilon.$$

Let $r$ denote a point on the interval $[0, a]$, where $p$ attains its maximum value. If $r$ is an interior point of $[0, a]$, then, clearly, $p'(r) = 0$. In the case $r = a$, we again conclude that $p'(r) = 0$, as

$$p(x) \leq 1 + \varepsilon = p(a)$$

for $x \in [a, b]$. In either case $r$ is a root of the polynomial $p'$ distinct from $x_1, \ldots, x_{n-1}$, so we have arrived at a contradiction with the fact that $\deg p' = 2n - 2$.

(iii) It is easy to see that $p''$ has a root $s_j$ on each open interval $(x_j, x_{j+1})$, $j = 1, \ldots, n-2$. Since $p''$ is odd it also has the roots $0, -s_1, \ldots, -s_{n-2}$. Clearly, since $\deg p'' \le 2n-3$, it does not have any other roots. Therefore, $p'$ is monotone on the interval $[0, x_1]$. If it increases on this interval, then it is negative there, as $p'(x_1) = 0$. So, in this case, $p$ decreases on the interval $[0, x_1]$, which contradicts the fact $p(a) > 0$. Thus, $p'$ decreases on the interval $[0, x_1]$. Moreover, it has a maximum at $x = 0$, for it is an even polynomial. Finally, there exists a point $x \in (0, a)$ such that

$$p'(x) = (1 - \varepsilon)/a,$$

since $p(0) = 0$ and $p(a) = 1 - \varepsilon$. Due to monotonicity of $p'$ we conclude

$$p'(0) \ge (1 - \varepsilon)/a.$$

(iv) Consider $t > 0$ and let $q(x) = p_{n,ta,tb}(tx)$. Also consider the points $y_0 = ta, y_1, \ldots, y_n = tb$ of the Chebyshev alternance for $p_{n,ta,tb} - 1$. It is easy to see that the points $y_0/t, y_1/t, \ldots, y_n/t$ constitute a Chebyshev alternance for $q - 1$ and by Theorem 1 we conclude that $q = p_{n,a,b}$. The equality $\varepsilon(n, ta, tb) = \varepsilon(n, a, b)$ easily follows. $\qquad\square$

## C   PROOF OF PROPOSITION 2

*Proof.* We denote $p_{2,a,b}$ and $\varepsilon(2, a, b)$ by $p$ and $\varepsilon$ respectively throughout this proof. From Proposition 1 we conclude that $p$ satisfies $p(a) = 1 - \varepsilon, p(e) = 1 + \varepsilon$, and $p(b) = 1 - \varepsilon$, where $e \in (a, b)$ and $\varepsilon = \|p - 1\|_{C[a,b]}$. Since $p'(e) = 0$ it is clear that $p'(x) = \alpha\left(e^2 - x^2\right)$ and, therefore, $p(x) = \alpha\left(e^2 x - x^3/3\right)$ for some $\alpha \in \mathbb{R}$. Now the equation $p(a) = p(b)$ implies $e^2(a - b) = \left(a^3 - b^3\right)/3$, so $e^2 = \left(a^2 + ab + b^2\right)/3$. That is, $p(x) = \alpha/3\left(\left(a^2 + ab + b^2\right)x - x^3\right)$ with some $\alpha \in \mathbb{R}$. In order to find $\alpha$ and $\varepsilon$ we calculate

$$1 - \varepsilon = p(a) = p(b) = \frac{\alpha}{3}\left(a^2 b + b^2 a\right), \, 1 + \varepsilon = p(e) = \frac{2\alpha}{3}\left(\frac{a^2 + ab + b^2}{3}\right)^{3/2}.$$

Thus,

$$\frac{\alpha}{3}\left(2\left(\frac{a^2 + ab + b^2}{3}\right)^{3/2} + a^2 b + b^2 a\right) = 2$$

$$\alpha = \frac{6}{2\left(\frac{a^2+ab+b^2}{3}\right)^{3/2} + a^2 b + b^2 a}$$

$$\varepsilon = \frac{\alpha}{6}\left(2\left(\frac{a^2 + ab + b^2}{3}\right)^{3/2} - a^2 b - b^2 a\right) = \frac{2\left(\frac{a^2+ab+b^2}{3}\right)^{3/2} - a^2 b - b^2 a}{2\left(\frac{a^2+ab+b^2}{3}\right)^{3/2} + a^2 b + b^2 a}$$

$\qquad\square$

## D   PROOF OF PROPOSITION 3

**Proposition 5.** *With the definitions 4 the sequences $a_n$ and $b_n$ converge to 1, and $b_n - a_n$ converges to zero quadratically. More precisely,*

$$\lim_{n \to \infty} \frac{b_{n+1} - a_{n+1}}{(b_n - a_n)^2} = \frac{3}{8}.$$

*Proof.* At first it is easy to see that $a_n + b_n = 2$ for all $n \in \mathbb{N}$. So, without loss of generality we can assume that $a + b = 2$. With this assumption we can rewrite the function $\varepsilon(2, \cdot, \cdot)$ in the following form

$$\varepsilon(2, a, b) = \frac{\left(\frac{4-ab}{3}\right)^{3/2} - ab}{\left(\frac{4-ab}{3}\right)^{3/2} + ab}.$$

Now we claim that $\varepsilon(2, a, b) < (b - a)/2$. This can be checked directly from the formula, but the implicit argument can be made based on the definition of $\varepsilon$. Since the polynomial $p(x) = x$ satisfies $\|p - 1\|_{C[a,b]} = (b - a)/2$ and $p$ is not optimal, we get that $\varepsilon(2, a, b) < (b - a)/2$. From this we conclude that $a_1 > a$ and $b_1 < b$. By induction we get that $\{a_n\}$ is increasing and $\{b_n\}$ is decreasing. Since also $a_n < b_n$ for all $n$ we obtain that these sequences converge to some points $A$ and $B$ respectively. Clearly, $a < A \leqslant B < b$ and $A + B = 2$. Since $\varepsilon$ is a continuous function, we can pass to the limit and obtain $A = 1 - \varepsilon(2, A, B)$ and $B = 1 + \varepsilon(2, A, B)$. Again, it can be checked directly that this implies $A = B$, but according to definition of $\varepsilon$ we get that provided $A < B, A = 1 - \varepsilon(2, A, B)$, and $B = 1 + \varepsilon(2, A, B)$ it follows that $p(x) = x$ is the best degree three odd polynomial approximation of unity of $[A, B]$, which is not true. Thus, $A = B = 1$ and it remains to prove the quadratic rate of convergence.

Using the assumption $a + b = 2$ we get that $ab = \left((a + b)^2 - (a - b)^2\right)/4 = 1 - (a - b)^2/4$. Now with this we calculate (we let $\gamma(a, b)$ denote the expression $\left(\frac{4-ab}{3}\right)^{3/2} + ab$)

$$b_1 - a_1 = 2\varepsilon(2, a, b) = 2\frac{\left(\frac{4-ab}{3}\right)^{3/2} - ab}{\left(\frac{4-ab}{3}\right)^{3/2} + ab} = 2\frac{\left(\frac{4-ab}{3}\right)^3 - a^2b^2}{\gamma(a,b)^2} = \frac{2}{27\gamma(a,b)^2}\left((4 - ab)^3 - 27a^2b^2\right) =$$

$$\frac{2}{27\gamma(a,b)^2}\left(64 - 48ab - 15a^2b^2 - a^3b^3\right) = \frac{2(b-a)^2}{27\gamma(a,b)^2}\left(\frac{81}{4} - \frac{9}{8}(b-a)^2 + \frac{(b-a)^4}{64}\right).$$

Since this calculation also works for $b_{n+1} - a_{n+1}$ and using that $a_n, b_n \to 1$ we get that

$$\frac{b_{n+1} - a_{n+1}}{(b_n - a_n)^2} = \frac{2}{27\gamma(a_n, b_n)^2}\left(\frac{81}{4} - \frac{9}{8}(b_n - a_n)^2 + \frac{(b_n - a_n)^4}{64}\right) \to \frac{3}{8}$$

as $\gamma(1, 1) = 2$. $\qquad\square$

Now we are ready to prove Proposition 3.

*Proof.* From the definition of $a_n, b_n$, it follows that

$$b_n - a_n = (1 - \varepsilon_n) - (1 + \varepsilon_n) = 2\varepsilon_n.$$

Using Proposition 5, we get

$$\frac{\varepsilon_{n+1}}{\varepsilon_n^2} = \frac{\frac{1}{2}(b_{n+1} - a_{n+1})}{\frac{1}{4}(b_n - a_n)^2} \to \frac{3}{4}.$$

From the proof of Proposition 5, we know that

$$\frac{b_{n+1} - a_{n+1}}{(b_n - a_n)^2} = \frac{2}{27\gamma(a_n, b_n)^2}\left(\frac{81}{4} - \frac{9}{8}(b_n - a_n)^2 + \frac{(b_n - a_n)^4}{64}\right) =$$

$$= \frac{2\left(\frac{81}{4} - \frac{9}{8}(2\varepsilon_n)^2 + \frac{1}{64}(2\varepsilon_n)^4\right)}{27\left(\left(\frac{4-(1-\varepsilon_n)(1+\varepsilon_n)}{3}\right)^{3/2} + (1 - \varepsilon_n)(1 + \varepsilon_n)\right)^2} =$$

$$= \frac{\left(\frac{81}{2} - 9\varepsilon_n^2 + \frac{1}{2}\varepsilon_n^4\right)}{27\left(\left(1 + \frac{\varepsilon_n^2}{3}\right)^{3/2} + 1 - \varepsilon_n^2\right)^2},$$

For $\varepsilon_n \in (0, 1)$, this expression is no more than $1/2$.

$$\frac{2\varepsilon_{n+1}}{4\varepsilon_n^2} = \frac{b_{n+1} - a_{n+1}}{(b_n - a_n)^2} \leq \frac{1}{2}$$

Therefore, $\varepsilon_{n+1} \leq \varepsilon_n^2$. $\qquad\square$

# E    PROOF OF COROLLARY 1

*Proof.* Let $[a_0, b_0] = [a_0, 1]$ be the starting segment, $0 < a_0 < 1$. From 3, we can write approximation error after the first iteration:

$$\varepsilon_0 = \frac{2\left(\frac{a_0^2+a_0+1}{3}\right)^{3/2} - a_0^2 - a_0}{2\left(\frac{a_0^2+a_0+1}{3}\right)^{3/2} + a_0^2 + a_0} = 1 - \frac{2a_0^2 + 2a_0}{2\left(\frac{a_0^2+a_0+1}{3}\right)^{3/2} + a_0^2 + a_0} < 1 - a_0.$$

After the first iteration, we start the recursion

$$a_{n+1} = 1 - \varepsilon_n, b_{n+1} = 1 + \varepsilon_n.$$

From Proposition 3, $\varepsilon_{n+1} \leq \varepsilon_n^2$ and by recursion we get

$$\varepsilon_n \leq \varepsilon_0^{2^n} \leq (1 - a_0)^{2^n}.$$

Then we can find the number of steps, necessary to get the desired error of approximation $\varepsilon$:

$$n \leq \left\lceil \log_2\left(\frac{\ln \varepsilon}{\ln(1 - a_0)}\right) \right\rceil.$$

$\square$

# F    REMEZ ALGORITHM

Let us describe the main idea of the Remez algorithm. Assume that we are given a set $\{x_1, \ldots, x_{n-1}\}$ of distinct points on the open interval $(a, b)$.

1. **Use $x_0 = a, x_1, \ldots, x_{n-1}, x_n = b$ as a guess for the Chebyshev alternance points for $p_{n,a,b} - 1$.** It is easy to see that there is a unique pair $(p, \varepsilon)$ such that $p \in L_n$ (that is, $p$ is odd and has degree $\leq 2n - 1$), $\varepsilon \in \mathbb{R}$, and $p(x_j) = 1 - (-1)^j \varepsilon$ for all $j = 0, 1, \ldots, n$. The equations $p(x_j) = 1 - (-1)^j \varepsilon$ for $j = 0, \ldots, n$ form a nonsingular system of linear equations in $n + 1$ unknowns, namely, $\varepsilon$ and coefficients of $p$. Thus, $p$ and $\varepsilon$ are, indeed, uniquely determined by the above conditions.

2. **Solve the system $p(x_j) = 1 - (-1)^j \varepsilon$, where $j = 0, \ldots, n$ to find $\varepsilon$ and coefficients of $p$.** Unfortunately, $x_0, \ldots, x_n$ may not constitute a Chebyshev alternance for $p - 1$, as $p$ is not guaranteed to satisfy $p([a, b]) \subset [1 - \varepsilon, 1 + \varepsilon]$. However, it is clear that $p$ has exactly $n - 1$ distinct extremal points $\{y_1, \ldots, y_{n-1}\}$ in the open interval $(a, b)$.

3. **Find the extremal points $\{y_1, \ldots, y_{n-1}\}$ of $|p - 1|$ in the interval $(a, b)$, where $p$ has discovered coefficients.** The collection of points $y_0 = a, y_1, \ldots, y_{n-1}, y_n = b$ (consisting of boundaries of the interval and extremal points of $p$) serves as a new guess for the Chebyshev alternance points for $p_{n,a,b} - 1$, and this guess is better than the previous.

4. **Repeat algorithm starting with $y_0 \ldots y_n$.** By repeating the above construction with points $y_1, \ldots, y_{n-1}$ instead of $x_1, \ldots, x_{n-1}$, we obtain a new pair $(q, \delta)$ with similar properties. By a fairly straightforward argument one can show that $\delta \geq \varepsilon$ and $\|q - 1\|_{C[a,b]} \leq \|p - 1\|_{C[a,b]}$. Iterating this process yields a sequence of polynomials that is guaranteed to converge to $p_{n,a,b}$.

The pseudocode is presented in Algorithm 3 below.

It should be noted that Remez algorithm is notorious for its instability when dealing with polynomials of sufficiently high degree. However, we have not observed an improvement of our methods when using polynomials of degrees higher than 5.

---

**Algorithm 3** Remez algorithm

---

**Require:** $n = (degree + 1)/2$, $a < b$, max_iterations $> 0$, tolerance
**Ensure:** Optimal polynomial $p \in L_n$ and error bound $\varepsilon$
   Initialize $x \leftarrow [x_0, x_1, \ldots, x_n]$ where $x_0 = a$, $x_n = b$
   iteration_count $\leftarrow 0$
   prev_epsilon $\leftarrow 0$
   **for** iteration_count $= 1 \ldots$ max_iterations **do**
      Construct $(n+1) \times (n+1)$ matrix $A$, where $A_{ij} = x_i^{2j+1}$ for $j = 0 \ldots n-1$, $A_{i,n} = (-1)^{i+1}$
      Construct right-hand side vector $b$, where $b_i = 1$
      solution $\leftarrow$ SolveLinearSystem$(A, b)$
      $p_{\text{coeffs}} \leftarrow$ solution$[0{:}n]$                     $\triangleright$ Polynomial coefficients
      $\varepsilon \leftarrow$ solution$[n]$                           $\triangleright$ Error parameter
      Find all local extrema $y_1, \ldots, y_{n-1}$ of $|p(x) - 1|$ in $(a, b)$
      Update points: $x \leftarrow [a, y_1, y_2, \ldots, y_{n-1}, b]$
      $\varepsilon_{new} \leftarrow \max_i(|p(y_i) - 1|)$                $\triangleright$ New error
      **if** $\varepsilon < \varepsilon_{new}$ + tolerance **then**
         **return** $(p, \varepsilon)$
   **return** $(p, \varepsilon)$

---

# G   Proof of Proposition 4

*Proof.* (i). $d \in \mathbb{N}, d \geq 2$ and consider the function $E(t) = \varepsilon(d, t, 1 + \delta)$. It is easy to see that $E$ is continuous, $E$ monotonically decreases on the interval $t \in (0, 1 + \delta)$ and satisfies $E(t) \to 1$ as $t \to 0$, and $E(t) \to 0$ as $t \to 1 + \delta$. Thus, there exists a unique $a = a(d, \delta) \in (0, 1 + \delta)$ such that $E(a) = \delta$. Note that $E(1 - \delta) < \delta$, as the polynomial $p(x) = x$ approximates unity with error $\delta$ on the interval $[1 - \delta, 1 + \delta]$, even though it is not optimal (since $d \geq 2$). Thus, the error of the best approximation on $[1 - \delta, 1 + \delta]$ has to be strictly less than $\delta$. Therefore, $E(1 - \delta) < \delta$, so $a(d, \delta) \in (0, 1 - \delta)$.

(ii) and (iii). Let $a$ denote the solution of the equation $\varepsilon(d, a, 1 + \delta) = \delta$ and consider the corresponding polynomial $q_{d,\delta} = p_{d,a,1+\delta}$. By definition $q_{d,\delta}(x) \in [1 - \delta, 1 + \delta]$ for $x \in [a, 1 + \delta]$. Moreover, from Proposition 1 (iii) it follows that $q_{d,\delta}$ is concave and increasing on the interval $[0, a]$, so from the fact $q_{d,\delta}(a) = 1 - \delta$ we derive the inequalities $1 - \delta \geq q_{d,\delta(x)} \geq (1 - \delta)x/a$ for $x \in [0, a]$. Thus, $q_{d,\delta} \in \mathcal{P}_{d,\delta}$. Note that, in particular, we have proved the inequality of (iii) for $q_{d,\delta}$. Now we prove that for all $p \in \mathcal{P}_{d,\delta}$ such that $p \neq q$ we have $\mathfrak{a}_\delta(p) > a$. From the definition of $\mathfrak{a}_\delta(p)$ we get that $\|p - 1\|_{C[\mathfrak{a}_\delta(p), 1+\delta]} \leq \delta$, hence, $E(\mathfrak{a}_\delta(p) = \varepsilon(d, \mathfrak{a}_\delta(p), 1 + \delta) \leq \delta$. Thus, by monotonicity of $E$ we infer that $\mathfrak{a}_\delta(p) \geq a$. If the equality $\mathfrak{a}_\delta(p) = a$ holds, then $p$ is an approximation of unity on $[a, 1 + \delta]$ with the error $\delta$, so it coincides with $q_{d,\delta}$ by the uniqueness of the best polynomial approximation. Otherwise, $\mathfrak{a}_\delta(p) > a$.

(iv). Let us state an *auxiliary fact*. Assume that polynomials $p, q \in L_d$ and points $0 < y_1 < y_2 < \cdots < y_d$ satisfy the inequalities $(-1)^{j-1}(q(y_j) - p(y_j)) \geq 0$ hold for all $j = 1, \ldots, d$. Then $q'(0) \geq p'(0)$. Assuming that this fact is true we can easily finish the proof. Indeed, assume that $x_0 = a(d, \delta) < x_1 < \cdots < x_d = 1 + \delta$ are the alternance points of $q_{d,\delta}$ and that $x_2 \geq 1 - \delta$. Now consider arbitrary $p \in \mathcal{P}_{d,\delta}$. We claim that $(-1)^{j-1}(q_{d,\delta}(x_j) - p(x_j)) \geq 0$ for all $j = 1, \ldots, d$. Indeed, if $j = 1$, then $q_{d,\delta}(x_1) = 1 + \delta \geq p(x_1)$ by definition of $\mathcal{P}_{d,\delta}$. If $j \geq 2$, then $x_j \geq 1 - \delta$ and the inequality holds since $q(x_j) = 1 - (-1)^j \delta$ and $|p(x_j) - 1| \leq \delta$. Thus, it remains to prove the foregoing auxiliary fact. Let us fix polynomials $p, q \in L_d$ and points $0 < y_1 < y_2 < \cdots < y_d$ such that the inequalities $(-1)^{j-1}(q(y_j) - p(y_j)) \geq 0$ hold for all $j = 1, \ldots, d$. Consider polynomials $\lambda_j \in L_d, j = 1, \ldots, n$ such that $l_j(x_k) = \delta_{jk}$, where $\delta_{jk}$ is the Kronecker's symbol. It is easy to verify that the polynomials $l_j$ indeed exist and are unique. Moreover, $p$ and $q$ can be recovered by an analog of the Lagrange's interpolation formula $p = \sum_{j=1}^d p(x_j)\lambda_j$ and $q = \sum_{j=1}^d q(x_j)\lambda_j$. Thus, $q'(0) - p'(0) = \sum_{j=1}^d (q(x_j) - p(x_j))\lambda_j'(0)$. The proof finishes by observing that $(-1)^{j-1}\lambda_j'(0) > 0$. To prove this observation note that all $2d - 1$ roots of $\lambda_j$ are simple and real. Therefore, the sign of the derivative $\lambda_j'$ alternates on the roots of $\lambda_j$ enumerated in increasing order. That is, in the vector

$$\begin{pmatrix} \lambda_j'(0) & \lambda_j'(x_1) & \ldots & \lambda_j'(x_{j-1}) & \lambda_j'(x_{j+1}) & \ldots & \lambda_j'(x_d) \end{pmatrix} \tag{10}$$

the signs of components are alternating. Finally, since $\lambda_j(x_j) = 1 > 0$ it follows that $\lambda'_j(x_{j-1}) \geq 0$ and $\lambda'_j(x_{j+1}) \leq 0$ (if $j = 1$ or $j = d$ only one of these inequalities should be stated). The inequality $(-1)^{j-1}\lambda'_j(0) > 0$ now easily follows from the alternating property of the vector equation 10.

$\square$

*Remark.*

1. The value $a(d, \delta)$ introduced in Proposition 4 (i) is given there as the solution of the equation $\varepsilon(d, a, 1 + \delta) = \delta$. This allows to evaluate $a(d, \delta)$ by using binary search (given any algorithm that computes the function $\varepsilon$), since the left part of this equation is a continuous and decreasing function of $a$.

2. From Proposition 4 (iv) it is easy to see that $q_{d,\delta}$ is the solution to the problem equation 5 for $d = 2$. For larger degrees this statement is no longer true in general. However, it stays true provided $\delta$ is large enough. For example, by calculating $q_{d,\delta}$ numerically we observed that the condition of Proposition 4 (iv) is satisfied for $d = 3, \delta \geq 0.073$ and $d = 4, \delta \geq 0.201$. In general, for each $d$ there exists $\delta_d \in (0, 1)$ such that $q_{d,\delta}$ is the solution to the problem equation 5 for $\delta \geq \delta_d$.

3. It is easy to derive the formula for the classical Newton-Schulz iterations from the polynomials $q_{d,\delta}$. Indeed, consider $d = 2$ and then pass to the limit $\delta \to 0$. Clearly, the polynomial $q_{2,\delta}(x)$ converges to $p(x)$ such that $p(1) = 1$ and $p'(1) = 0$. There is only one odd polynomial of degree three satisfying these properties, namely, $p(x) = 3x/2 - x^3/2$, which is used in the classical Newton-Schulz iterations.

# H    EXPERIMENTAL DETAILS

NanoGPT (Jordan et al., 2024a) is trained on a subset of 0.8B training tokens of FineWeb dataset (Penedo et al., 2024) for 6200 steps with initial learning rate 0.0036 and trapezoidal schedule (1800 warmdown steps) on 1 A100 GPU. For normalization in our method, we used Gelfand's formula. For normalization in original Muon optimizer, Frobenius norm was used.

In practice, we have not observed any noticeable difference in runtime of Muon with different polynomials in experiment with NanoGPT. Each training step required 2.5-2.9 seconds for different polynomials. Theoretically this can be explained as follows. The FLOP overhead of Muon over SGD is $(T/3)m/B$ (see runtime analysis in (Jordan et al., 2024b)), where $m$ is matrix dimension, $B$ - sequence length, by $T$ we will denote number of matmuls ($T = 15$ for original Muon). The difference in overhead of Muon with polynomials with $T_1$ and $T_2$ matmuls is $((T_1 - T_2)/3)m/B$. In our experiment with NanoGPT, m=768, B=524288, the difference with original Muon is $T_1 - T_2 \leq 3$ so overhead is $((T_1 - T_2)/3)m/B \leq 0.0015$.

For training Wide ResNet-16-10 on CIFAR-10 with Riemannian SGD and ADAM, the learning rate is set to 0.2 and 0.4 for parameters restricted to Stiefel manifold and 0.01 otherwise. For standard SGD and ADAM learning rate is set to 0.1 and 0.0003 respectively. The experiments were run on 1 V100 GPU. For CANS retraction, one iteration of Algorithm 1 was enough in practice to perform orthogonalization.

# I    RIEMANNIAN SGD AND ADAM ON STIEFEL MANIFOLD

Table 2 shows results of training Wide ResNet-16-10 on CIFAR-10 with SGD on Stiefel manifold.

Algorithms 4 and 5 present Riemannian SGD and Adam on Stiefel manifold. Algorithm 6 presents algorithm of performing Cayley retraction using Woodbury formula.

Table 2: Accuracy and training time for Wide ResNet-16-10 on CIFAR-10 using SGD.

| Retraction | Accuracy | Time per epoch (s) |
| --- | --- | --- |
| - | **95.97** | **34.9** |
| Cayley (Li et al., 2020) | 94.81 | 69.5 |
| Cayley (Woodbury) | 94.93 | 68.8 |
| QR | 94.80 | 61.0 |
| CANS | 94.73 | 43.6 |

---

**Algorithm 4** SGD with momentum on Stiefel manifold

---

**Input** Momentum $\beta$, learning rate $\alpha$.
Initialize $X_1 \in \mathbb{R}^{n \times p}$ as orthonormal matrix.
**for** $1 \ldots \texttt{n\_iters}$ **do**
$\quad M_{k+1} = \beta M_k - G(X_k)$
$\quad M_{k+1} = M_{k+1} - \frac{1}{2} X_k(M_{k+1}^T X_k + X_k^T M_{k+1})$
$\quad X_{k+1} = \text{Retr}(X_k + \alpha M_{k+1})$

---

**Algorithm 5** Adam on Stiefel manifold

---

**Input** Momentum coefficients $\beta_1, \beta_2$, learning rate $\alpha$.
Initialize $X_1 \in \mathbb{R}^{n \times p}$ as orthonormal matrix.
**for** $k$ in $1 \ldots \texttt{n\_iters}$ **do**
$\quad v_{k+1} = \beta_2 v_k + (1 - \beta_2)\|G(X_k)\|_F^2$
$\quad \hat{v}_{k+1} = v_{k+1}/(1 - \beta_2^k)$
$\quad M_{k+1} = \beta_1 M_k - (1 - \beta_1)G(X_k)$
$\quad \hat{M}_{k+1} = M_{k+1}/(1 - \beta_1^k)$
$\quad \hat{M}_{k+1} = \hat{M}_{k+1} - \frac{1}{2} X_k(\hat{M}_{k+1}^T X_k + X_k^T \hat{M}_{k+1})$
$\quad X_{k+1} = \text{Retr}(X_k - \alpha \hat{M}_{k+1}/\sqrt{\hat{v}_{k+1} + \epsilon})$
$\quad M_{k+1} = (1 - \beta_1^k)\hat{M}_{k+1}$

---

**Algorithm 6** Cayley retraction via Woodbury formula

---

**Input** Parameters $X_k$, step direction $M_{k+1}$, learning rate $\alpha$.
$L = [\alpha M_{k+1}; X_k]$
$R = \begin{bmatrix} X_k^T \\ \alpha(M_{k+1}^T X_k X_k^T - M_{k+1}^T) \end{bmatrix}$
$Y = X_k + \frac{1}{2}\alpha M_{k+1}$
$X_{k+1} = Y + \frac{1}{2}L(I - \frac{1}{2}RL)^{-1}RY$
**Return:** $X_{k+1} = CayleyRetr(X_k + \alpha M_{k+1})$

---

## J  POLYNOMIALS

Coefficients are presented from left to right from the minimal degree to maximum. For example, for coefficients [(a, b), (c, d, e)] the composition is $p_2(p_1(x))$, where $p_1(x) = ax + bx^3$, $p_2(x) = cx + dx^3 + ex^5$.

Original Muon coefficients of 3-rd order polynomial for any number of iterations: [(3.4445, -4.7750, 2.0315)]*num_iters (green in Figure 3, 5).

CANS, eps=0.3, order=3, iter=7, mm=14 (black in Figure 3)
[(5.181702879894027, −5.177039351076183),
(2.5854225645668487, −0.6478627820075661),
(2.565592012027513, −0.6452645701961278),
(2.5162233474315263, −0.6387826202434335),
(2.401068707564606, −0.6235851252726741),
(2.1708447617901196, −0.5928497805346629),
(1.8394377168195162, −0.5476683622291173)]

CANS, eps=0.3, order=5, iter=5, mm=15 (purple in Figure 3)
[(8.492217149995927, −25.194520609944842, 18.698048862325017),
4.219515965675824, −3.1341586924049167, 0.5835102469062495),
(4.102486923388631, −3.0527342942729288, 0.5742243021935801),
(3.68500495227776493, −2.756862315006488, 0.5405198817097779),
2.734387280007103, −2.036641382834855, 0.4592314693659632)]

CANS, eps=0.00188, order=3, iter=9, mm=18 (purple in Figure 4)
[(5.179622107852338, −5.174287102735334),
(2.5836099434139492, −0.6476254200945953),
(2.5610021062961206, −0.6446627537769272),
(2.505058237036672, −0.6373139418181356),
(2.3764825571306125, −0.6203257475007262),
(2.1279007426858794, −0.5870609391939776),
(1.7930526112541054, −0.5412446350453286),
(1.5582262242936464, −0.5082920767544266),
(1.5021988305175455, −0.5003140810786916)]

CANS, eps=0.00443, order=3, iter=9, mm=18 (blue in Figure 4)
[(5.182503604966906, −5.178098480082684),
(2.586120737395915, −0.6479542005271643),
(2.567364126726186, −0.6454968804392178),
(2.520560084348265, −0.6393528082067044),
(2.410759275435182, −0.6248683598710716),
(2.1883348130094173, −0.5952022073798908),
(1.8595760874873613, −0.5504490972723968),
(1.589020160467417, −0.5126569802066718),
(1.5051653981684994, −0.5007377068751799)]

CANS, eps=0.0035, order=3, iter=9, mm=18 (grey in Figure 5)
[(5.181724335835382, −5.177067731075524),
(2.585441267930541, −0.6478652310697918),
(2.5656394547047783, −0.6452707898813249),
(2.5163392603382473, −0.6387978622974516),
(2.401326686185833, −0.6236192975654269),
(2.17130618635129, −0.5929118810597139),
(1.8399595521688579, −0.5477404797274893),
(1.5792011481985957, −0.5112666878668612),
(1.5040821254913361, −0.500583031372834)]

CANS, eps=0.3, order=5, iter=4, mm=12 (purple in Figure 5)
[(8.420293602126344, −24.910491192120688, 18.472094206318726),
(4.101228661246281, −3.0518555467946813, 0.5741241025302702),

$(3.6809819251109155, -2.75396502307162, 0.5401902781108926),$
$(2.7280916801566666, -2.0315492757300913, 0.45866431681858805)]$

Jiacheng's, order=5, iter=6, mm=18 (green in Figure 4)
$[(3955/1024, -8306/1024, 5008/1024),$
$(3735/1024, -6681/1024, 3463/1024),$
$(3799/1024, -6499/1024, 3211/1024),$
$(4019/1024, -6385/1024, 2906/1024),$
$(2677/1024, -3029/1024, 1162/1024),$
$(2172/1024, -1833/1024, 682/1024)]$

Jiacheng's, order=5, iter=5, mm=18
$[(3839/1024, -8060/1024, 4883/1024),$
$(3851/1024, -7277/1024, 3966/1024),$
$(4011/1024, -6812/1024, 3318/1024),$
$(2738/1024, -3261/1024, 1321/1024),$
$(2172/1024, -1833/1024, 683/1024)]$

## K    TIME

The number of matmuls is proportional to FLOPS and to the spent time up to the errors. Table 3 below shows time for Figure 2 (on CPU in seconds).

Table 3:  Time for matrix orthogonalization in Figure 1 (on CPU in seconds).

| Method | Matmuls | Time |
|---|---|---|
| classic Newton-Schultz | 60 | 6.57 |
| 3-rd order | 26 | 2.70 |
| 5-th order | 24 | 1.96 |
| classic Newton-Schultz, Gelfand | 60 | 6.57 |
| 3-rd order, Gelfand, $a_0 = 1e - 3$ | 32 | 3.27 |
| 5-th order, Gelfand, $a_0 = 1e - 3$ | 30 | 2.79 |
| 3-rd order, Gelfand, $a_0 = 1e - 7$ | 44 | 4.72 |
| 5-th order, Gelfand, $a_0 = 1e - 7$ | 42 | 3.80 |

## L    ABLATION OF MATRIX NORMALIZATION

We compare the effect of normalization before orthogonalization in the Muon optimizer. Figure 6 shows that Muon with Gelfand's normalization has improved convergence.

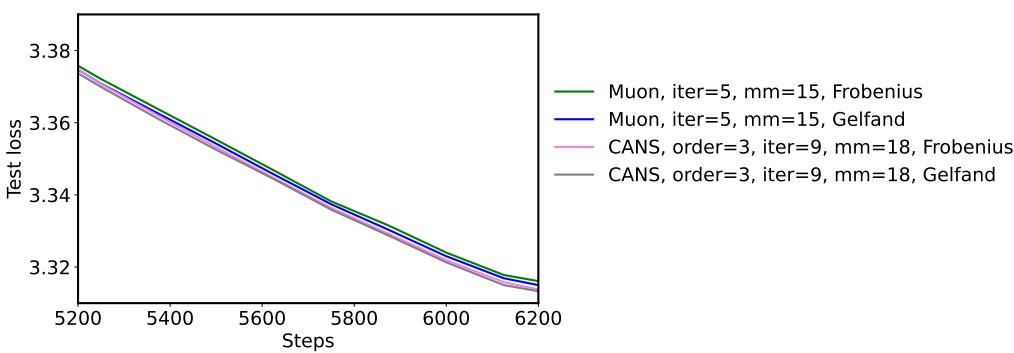

Figure 6:  NanoGPT test loss curves for Muon with Gelfand's and Frobenius normalization before orthogonalization.

Figures 7 and 9 show the full training and test loss curves of NanoGPT.

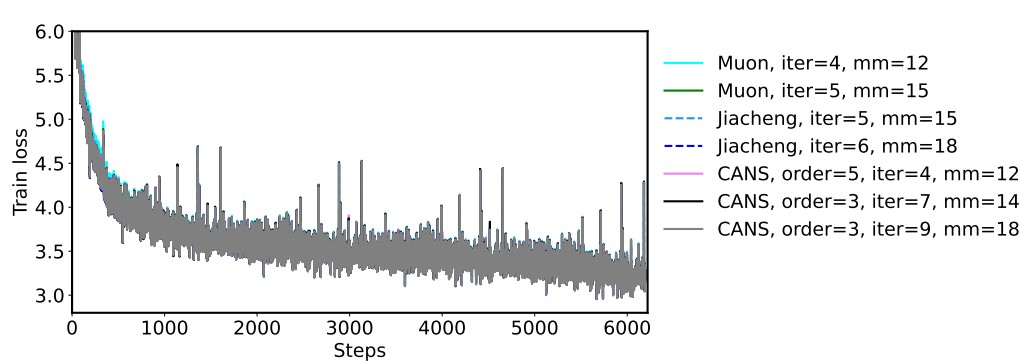

Figure 7: NanoGPT full train loss curve.

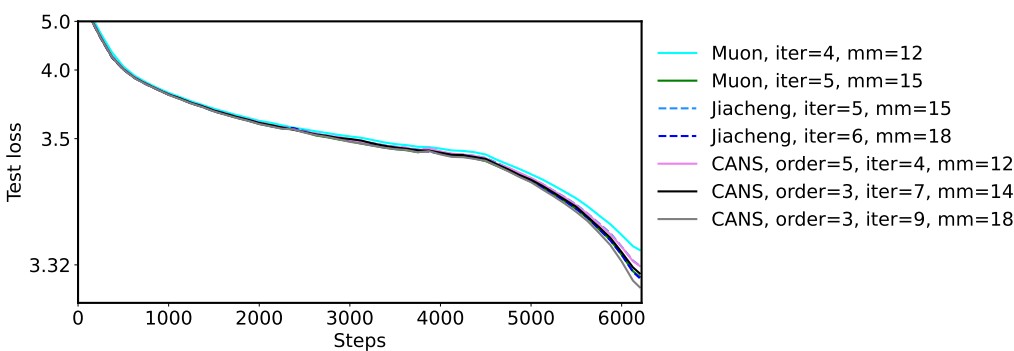

Figure 8: NanoGPT full test loss curve.

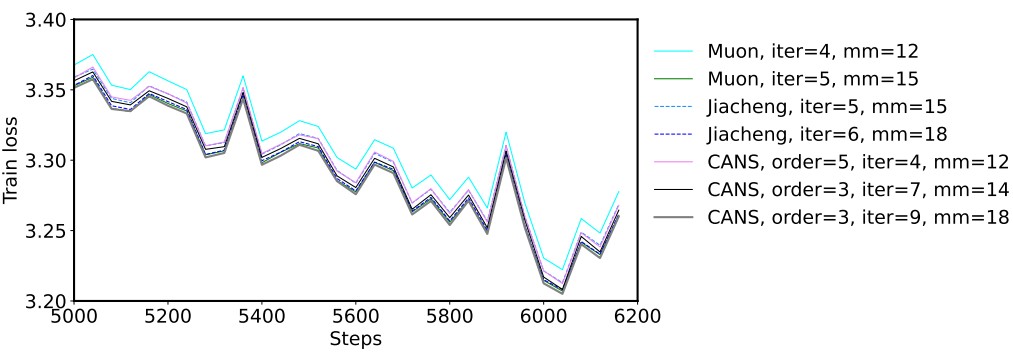

Figure 9: NanoGPT smoothed train loss curve.

Table 4: Time for retraction of $n \times p$ matrix.

| n | p | Cayley | QR | CANS |
|---|---|---|---|---|
| 1024 | 32 | 0.11 | 0.28 | 0.07 |
| 1024 | 64 | 0.13 | 0.47 | 0.07 |
| 1024 | 128 | 0.19 | 0.86 | 0.08 |
| 1024 | 256 | 0.28 | 1.83 | 0.11 |
| 1024 | 512 | 0.43 | 3.55 | 0.23 |
| 1024 | 1024 | 0.70 | 6.61 | 0.59 |
| 2048 | 32 | 0.22 | 0.32 | 0.07 |
| 2048 | 64 | 0.29 | 0.54 | 0.08 |
| 2048 | 256 | 0.77 | 2.35 | 0.15 |
| 2048 | 512 | 1.33 | 4.54 | 0.43 |
| 2048 | 1024 | 2.53 | 9.08 | 1.11 |
| 2048 | 2048 | 4.98 | 18.03 | 3.99 |
| 4096 | 32 | 0.68 | 0.48 | 0.08 |
| 4096 | 64 | 0.96 | 0.89 | 0.09 |
| 4096 | 512 | 5.08 | 7.99 | 0.71 |
| 4096 | 1024 | 9.74 | 15.84 | 2.13 |
| 4096 | 2048 | 18.89 | 34.02 | 8.20 |
| 4096 | 4096 | 37.04 | 68.19 | 30.57 |
| 8192 | 32 | 2.46 | 0.67 | 0.08 |
| 8192 | 64 | 3.59 | 1.30 | 0.10 |
| 8192 | 1024 | 37.42 | 24.40 | 4.20 |
| 8192 | 2048 | 73.94 | 55.65 | 16.64 |
| 8192 | 4096 | 145.71 | 130.80 | 62.68 |
| 8192 | 8192 | 290.25 | 321.01 | 236.78 |

## M  TIME COMPARISON OF RETRACTIONS

Table 4 shows time (in seconds) for retraction of $n \times p$ matrix measured on A100. For a small step-size, it is enough to make 2 iterations of Cayley or 1 CANS iteration to reach nearly the same desired accuracy of orthogonalization.

