# OpenReview forum: "Accelerating Newton-Schulz Iteration for Orthogonalization via Chebyshev-type Polynomials"
_ICLR.cc/2026/Conference — Submitted to ICLR 2026_

### Official Review · Reviewer_6mHR · 2025-10-22

**Soundness:** 3
**Presentation:** 2
**Contribution:** 3
**Rating:** 4
**Confidence:** 2

**Summary:**

This paper introduces the 3rd order Newton-Schulz method with optimal coefficients through Chebyshev’s alternance theorem, and higher order extension via Remez algorithm. The authors then apply it to Muon optimizer and to Riemannina optimization on the Stiefel manifold, and shows acceleration from the proposed algorithms on both cases.

**Strengths:**

This paper improves the classic Newton-Schultz algorithm and shows acceleration on Muon and Riemannian optimization settings both theoretically and empirically.

**Weaknesses:**

The paper reads more like a mathematics paper than a machine learning paper. It lacks high-level intuition and is overly detailed, making it difficult to follow. The presentation is not very well-organized.

**Questions:**

1.In the Muon application, only partial test loss results are shown. It would be helpful to also include training loss curves and full test loss trajectories to verify the theory. Also, please clarify what “optimal” means — is it optimal in the sense of training (optimization) performance?

2.In section 4,

2.1 Equations (5) and (6) are unclear and should be rewritten more rigorously.

2.2 There are two “Algorithm 1” entries — please correct accordingly.

2.3 Lines 276–285 are difficult to interpret. It seems you are discussing an extension that uses different polynomials at each iteration. This should either be formalized as a theorem, lemma, or corollary, or rewritten more intuitively as a high-level remark rather than a sequence of technical statements.

2.4 How are d_i chosen at each step?

3.What is the relationship between 1-\delta and a? According to Proposition 4, a does not seem to be symmetric w.r.t. 1

4.Please introduce Remez algorithm in the corresponding part. It’s unclear to me how the coefficients of higher-order (>3) polynomials are chosen (as well as the order in the above question).

5.Please introduce the relationship Muon algorithm and Newton-Schultz more formally (at least you should state the whole algorithm of Muon), and then compare the corresponding part with your proposed optimal algorithm.

6.In Figure 3 and 4, the authors show the comparison between their proposed polynomials and original Muon. It seems that whether the polynomial is closer to 1 is not important since Muon has smaller right end point and are closer to 1 in most of the points than the proposed polynomials, and only the derivative at zero matters. How should this be understood?


Minor:

Line184: “greater or equal than”

Line 238: “On the other hand”

---

> ### Author Response · Authors · 2025-11-21
>
> Dear Reviewer 6mHR, thank you for your questions! We will clarify the details of our approach.
>
> W1. We thank the reviewer for the feedback on the paper's presentation. We will be sure to carefully revise our final version of the manuscript to provide more high-level intuition, making the core ideas more accessible to a general audience without compromising on the technical rigor of our contribution.
>
> 1. We have added the full train and test loss curves to Appendix L. Unfortunately, the differences are not clearly visible on this scale.
> The polynomials are optimal in the sense of Chebyshev's theorem, meaning they minimize the approximation error to $f \equiv 1$.
>
>
> 2.1 We have fixed the notation in equations (5) and (6).
> Conceptually, the problem of finding a polynomial with a high derivative at zero (equation 5) is equivalent to minimizing the left endpoint $a$ of the interval $[a, b]$ subject to the constraint that the polynomial's deviation from $1$ does not exceed $\delta$ (equation 6).
>
> 2.2 Thank you for noting this typo, we have corrected it.
>
> 2.3 Thank you for your feedback. We see that this section has caused a misunderstanding among the reviewers and we have improved its readability and clarity in the revised text.
>
> 2.4 The degrees of the polynomials ($2d_i-1$) should be chosen beforehand by the allocated budget of matrix multiplications. Empirically, we observed no significant improvement from using polynomials of degree higher than $5$. Therefore, we recommend using polynomials of degree $3$ or $5$.
>
> 3. The optimal value of the left boundary satisfies $a \leq 1-\delta$.
>
> 4. We have added pseudocode for the Remez algorithm to the section F in Appendix. We will be glad to answer if you have any questions.
>
>
> 5. The step of Muon optimizer can be written as follows:
> $$M_k=\beta M_{k-1} +(1-\beta)G_k,$$
> $$W_k=W_{k-1}-\eta Ortho(M_k),$$
> where $W$ are the parameters that we wish to update, $\eta$ is the learning rate, $G_k$ is the gradient on the current step $k$, $M_k$ is the momentum on the current step, $Ortho(M_k)=argmin_{O} \{\|M_k-O\|_F:  O^TO=I \, \text{or} \, OO^T=I\}$. The authors of Muon proposed to perform orthogonalization operation $Ortho$ using Newton-Schulz iteration with specific polynomial, shown in Figure 3. In our work, we propose to substitute this hand-crafted polynomial with CANS polynomials, that have a higher derivative at zero or make the matrix more orthogonal. So we only change $Ortho$ operation in Muon and compare the results. We have added this clarification to the text.
>
> 6. We pay attention to 2 features of polynomials: the closeness of polynomials to 1 and their derivative at zero. Apparently, we do not need to be very close to $1$ to make the method work. For example, the authors of Muon were satisfied with the interval $[0.7, 1.3]$ (endpoint of this polynomial is indeed approximately 0.7). However, there is certainly a tradeoff between the closeness to 1 and the derivative at 0: the closer the polynomial is to $1$, the smaller the derivative and vice versa. By constructing optimal polynomials, we improve this tradeoff, allowing for polynomials that, e.g., have larger derivatives at 0, while still remaining in the interval $[0.7, 1.3]$.
>
> Our analysis focuses on two key properties of the polynomials: the proximity of polynomials to 1 and their derivative at zero. Notably, highly accurate approximation of $1$ is not strictly necessary for the Muon to be effective; for instance, the authors of Muon found the interval $[0.7, 1.3]$ to be sufficient (the polynomial's left endpoint is indeed approximately $0.7$). However, a fundamental trade-off exists between these two properties: a closer approximation to $1$ typically results in a smaller derivative at zero, which slows down the convergence, and vice versa. By constructing optimal polynomials, we improve upon this trade-off, enabling, for example, polynomials with larger derivatives at zero while still satisfying the same interval constraint $[0.7, 1.3]$.

---

> > ### Comment · Reviewer_6mHR · 2025-11-26
> >
> > I would like to thank the authors for the reply and revision. I am still not convinced of the experiments on comparing Muon and other optimizers. The theory should guarantee advantages on "optimization", but the curves do not show this advantage. I understand that the test performance is slightly better, but this is another perspective that is not the focus of the theory.

---

> > > ### Author Response · Authors · 2025-11-27
> > >
> > > We thank the reviewer for raising this important point. You are correct to highlight the distinction between optimization properties (training loss) and generalization (test performance). We also agree that the noise in the training loss curves of Figure 7 makes it challenging to see the differences. To address this, we have added a new figure (Figure 9) presenting a smoothed version of these curves to better visualize the underlying trends.

---

### Official Review · Reviewer_xmCb · 2025-10-28

**Soundness:** 3
**Presentation:** 3
**Contribution:** 2
**Rating:** 6
**Confidence:** 4

**Summary:**

This paper introduces the Chebyshev-Accelerated Newton-Schulz (CANS) framework to address the notable drawbacks of conventional matrix orthogonalization methods. The framework derives explicit formulas and analyzes the convergence of optimal odd polynomials, utilizing the Remez algorithm to compute their higher-order counterparts. Compared to classical methods like the Newton-Schulz iteration and Cayley retraction, the polynomials generated by CANS reduce computational complexity and accelerate convergence while maintaining accuracy. Furthermore, the authors validate the method's effectiveness through tasks such as accelerating parameter orthogonalization with the Muon optimizer, training NanoGPT, and performing fast retractions on the Stiefel manifold.

**Strengths:**

1.This paper systematically applies Chebyshev approximation theory to optimize the coefficients of the Newton-Schulz iteration, proposing the Chebyshev-Accelerated Newton-Schulz (CANS) framework for finding "provably optimal" odd polynomials.

2.The paper is built upon a solid mathematical theory, with detailed and rigorous proofs for each proposition and corollary provided in the appendix. This offers robust mathematical support for the uniqueness and key properties of the optimal odd polynomials.

3.The overall structure of the paper is clear. It begins by stating the problem and progressively develops its main theory, supplemented by figures and textual explanations to help readers understand the meaning behind each proposition.

**Weaknesses:**

1.The typesetting for the proof of Proposition 1 in the appendix is slightly disorganized and could be improved. Additionally, the paper contains some errors; for instance, in the equation on line 50, the final exponent appears to be a transpose symbol 'T' when it should likely be 't'. The paper also mistakenly includes two distinct algorithms both labeled as "Algorithm 1".

2.The experiments on the Stiefel manifold are conducted solely with a Wide ResNet-16-10 on the CIFAR-10 dataset. The evaluation does not cover other architectures  or larger datasets. This limited scope makes it difficult to assess the generalizability of the CANS retraction for deeper networks and larger-scale data, indicating a need for more extensive experiments.

**Questions:**

Regarding the core parameter $\delta$, the paper only sets specific values experimentally and does not provide a quantitative guide for its selection. It fails to clarify how one might determine an appropriate δ based on the singular value distribution of a matrix or the specific requirements of a task.

---

> ### Author Response · Authors · 2025-11-21
>
> Dear Reviewer xmCb, thank you for your questions and comments.
>
> W1. Thank you for noting these typos, we have corrected them in the revised version. We have refined the proof of Proposition 1, and we will continue to work on improving readability of other proofs in the final version.
>
> W2. Thank you for this comment, we agree that exploring diverse applications is valuable. In our initial submission, we focused on the Wide Resnet-16-10 / CIFAR-10 setup to ensure a direct and fair comparison with the established Cayley retraction, following the benchmark from [1].
>
> Regarding generalizability, we would like to emphasize that CANS is a drop-in replacement for any retraction on the Stiefel manifold. Its formulation does not depend on a specific architecture or dataset. To substantiate its broad applicability, we highlight two points:
>
> 1. The core advantage of CANS is its superior scaling for large, thin matrices, which are common in deep learning (e.g., in LoRA). The table below demonstrates that CANS is faster than both Cayley and QR retractions on large-scale matrices, making it particularly suitable for modern large-scale models.
>
> 2. Relevance to state-of-the-art applications: Optimization on the Stiefel manifold is actively used in many impactful areas, including learning orthogonal matrices that improve model quantization [2,3], training factors of LoRA  with orthogonal constraints [4, 5], and may also be used for training orthogonal transformers [6]. CANS provides a more efficient computational primitive for all these applications without altering their underlying optimization objectives.
>
>
> The table shows time (in seconds) for retraction of $n\times p$ matrix measured on A100. For a small step-size, it is enough to make 2 iterations of Cayley or 1 CANS iteration to reach nearly the same desired accuracy of orthogonalization.
>
> \begin{array}{ll|lll}
> n & p & Cayley & QR & CANS \\\\
> \hline
> 1024 & 32 & 0.11 & 0.28 & 0.07 \\\\
> 1024 & 64 & 0.13 & 0.47 & 0.07 \\\\
> 1024 & 128 & 0.19 & 0.86 & 0.08 \\\\
> 1024 & 256 & 0.28 & 1.83 & 0.11 \\\\
> 1024 & 512 & 0.43 & 3.55 & 0.23 \\\\
> 1024 & 1024 & 0.70 & 6.61 & 0.59 \\\\
> \hline
> 2048 & 32 & 0.22 & 0.32 & 0.07 \\\\
> 2048 & 64 & 0.29 & 0.54 & 0.08 \\\\
> 2048 & 256 & 0.77 & 2.35 & 0.15 \\\\
> 2048 & 512 & 1.33 & 4.54 & 0.43 \\\\
> 2048 & 1024 & 2.53 & 9.08 & 1.11 \\\\
> 2048 & 2048 & 4.98 & 18.03 & 3.99 \\\\
> \hline
> 4096 & 32 & 0.68 & 0.48 & 0.08 \\\\
> 4096 & 64 & 0.96 & 0.89 & 0.09 \\\\
> 4096 & 512 & 5.08 & 7.99 & 0.71 \\\\
> 4096 & 1024 & 9.74 & 15.84 & 2.13 \\\\
> 4096 & 2048 & 18.89 & 34.02 & 8.20 \\\\
> 4096 & 4096 & 37.04 & 68.19 & 30.57 \\\\
> \hline
> 8192 & 32 & 2.46 & 0.67 & 0.08 \\\\
> 8192 & 64 & 3.59 & 1.30 & 0.10 \\\\
> 8192 & 1024 & 37.42 & 24.40 & 4.20 \\\\
> 8192 & 2048 & 73.94 & 55.65 & 16.64 \\\\
> 8192 & 4096 & 145.71 & 130.80 & 62.68 \\\\
> 8192 & 8192 & 290.25 & 321.01 & 236.78 \\\\
> \end{array}
>
> Q. If the $\delta$-orthogonalization algorithm is used for preprocessing the matrix before exact orthogonalization to increase its smallest singular values, then it is better to take $\delta$ as close to 1 as possible. Larger $\delta$ will allow polynomials to have higher derivative and increase the smallest singular values faster. However, $\delta$ should not be too close to 1 to avoid numerical errors. In our experiment, which is shown in Figure 2, we have chosen $\delta=0.99$ for these reasons.
>
> If it is needed to construct a sequence of polynomials that fit into desired interval $[1-e, 1+e]$ (desired $e$ is known) and have a high derivative at zero (like in Muon optimizer), then the first option is to set $\delta=e$ and use the one found polynomial on each iteration. The second option is to vary $\delta$ on each iteration. This process is described in section 4 after Proposition 4. First, set $\delta=e$. Then find a larger $\delta_1$, such that polynomial $p_1$ maps $[1-\delta_1, 1+\delta_1]$ into $[1-\delta, 1+\delta]$ (may use binary search). Repeat this procedure several times, finding an increasing sequence $\delta_1, \delta_2 ...\delta_l$ and composition of corresponding polynomials $p_1(p_2(...p_l(x)...))$.
>
>
> [1] Li, Jun, Li Fuxin, and Sinisa Todorovic. "Efficient riemannian optimization on the stiefel manifold via the cayley transform."
>
> [2] Liu, Zechun, et al. "Spinquant: Llm quantization with learned rotations." arXiv preprint arXiv:2405.16406 (2024).
>
> [3] Akhondzadeh, Mohammad Sadegh, et al. "KurTail: Kurtosis-based LLM Quantization." arXiv preprint arXiv:2503.01483 (2025).
>
>
> [4] Park, Juneyoung, et al. "Riemannian Optimization for LoRA on the Stiefel Manifold." arXiv preprint arXiv:2508.17901 (2025).
>
> [5] Li, Zhizhong, et al. "StelLA: Subspace Learning in Low-rank Adaptation using Stiefel Manifold." arXiv preprint arXiv:2510.01938 (2025).
>
> [6] Zhang, Aston, et al. "On orthogonality constraints for transformers." Vol. 2. Association for Computational Linguistics, 2021.

---

### Official Review · Reviewer_H3Bj · 2025-10-30

**Soundness:** 3
**Presentation:** 2
**Contribution:** 2
**Rating:** 4
**Confidence:** 3

**Summary:**

This paper addresses the problem of computing optimal orthogonal approximations to matrices, a fundamental operation in machine learning applications such as the Muon optimizer and Riemannian optimization on the Stiefel manifold. The authors propose CANS (Chebyshev-accelerated Newton-Schulz), which optimizes the coefficients of Newton-Schulz iterations using Chebyshev's alternance theorem. For degree-3 polynomials, they derive explicit optimal formulas, and for higher degrees, they apply the Remez algorithm. The method is demonstrated on three applications: matrix orthogonalization, the Muon optimizer for neural network training, and retraction on the Stiefel manifold for Riemannian optimization.

**Strengths:**

1. The work presents a novel theoretical framework for optimizing Newton-Schulz iteration coefficients. While polynomial approximation theory is classical, its systematic application to this problem through Chebyshev's alternance theorem is creative.

2. The theoretical contributions are rigorous with complete proofs in the appendices. Proposition 2 provides closed-form solutions for degree-3 polynomials, and the convergence analysis establishing quadratic convergence is solid. The experimental validation spans multiple domains with appropriate baselines.

3. The paper is generally well-written with clear motivation.

**Weaknesses:**

1. The paper cites concurrent work by Amsel et al. (2025). A slightly more detailed comparison in the related work section could help readers more clearly understand the overlapping and distinct contributions of the two papers regarding the exact case.

**Questions:**

1. Section 3.3 states that Gelfand's formula to estimate $\sigma_1$ can be implemented "without introducing any extra matmuls". However, normalization of matrix is applized before NS iteration, does that mean $(A^T A)^k $ must be saved? Clarifying this is important for understanding the practical implementation overhead.

2. Regarding $\delta$-orthogonalization,

(1) could you elaborate more on why "If we use distinct polynomials on each iteration, we can achieve more rapid convergence"? Say, given arbitrary $\delta \in (0,1)$, since optimal $a = a(d, \delta) \in (0,1-\delta)$, could we instead do binary search for $a(d, \delta)$ in region $(0, 1-\delta)$ and iterates the process $p, \epsilon = remez(a, 1+\delta, 2d-1)$ until $|\epsilon - \delta| \le 1e-7$ as follows:

$L = 0; R = 1-\delta$
repeat:
  a = (L+R)/2

  p, ε = remez(a, 1+$\delta$, 2d-1)

  if $|\epsilon-\delta| ≤ tol$: break

  if $\epsilon$ > $\delta$: L = a

  else:      R = a


(2) Why we need rapid convergence for $\delta$-orthogonalization? Fast convergence is desirable for exact orthogonalization through optimal odd polynomials. However, the goal of $\delta$-orthogonalization is to a find a polynomial to push singular values into interval $[1-\delta, 1+\delta]$ while with large derivative at the origin $0$. However, the rapid convergence through distinct polynomials over iterations would easily make $\epsilon$ approaching 0 (as stated in proposition 3), which could easily make $\epsilon < \delta$, especially for high tolerance.

3. A high-level question is do we really need optimal/exact orthogonalization in real world applications like Muon optimizer?

---

> ### Author Response · Authors · 2025-11-21
>
> Dear Reviewer H3Bj, thank you for your feedback and comments.
>
> W1. We appreciate your feedback and think that both papers offer valuable perspectives on Newton-Schulz iteration.
> In a nutshell, the key distinction in  contributions for the exact case is that we propose a preprocessing step that pushes small singular values, and also the Gelfand normalization, which both speeds up the convergence in the exact case.
> The theory for third-degree polynomials is equivalent to ours. While [Amsel et al.] prove the optimality of their polynomial composition in the exact case, they fully omit the inexact case, which is the primary focus of our work.
> For more detail, see the updated Related work section.
>
>
> Q1. Thank you for this question. Fortunately, we can do a trick to avoid extra matmul operations. Indeed, for example, for 3rd degree polynomial, we do not need to save extra matrices.
> For higher degree, we can do a similar trick by saving intermediate powers if needed. See the Section 3.3 for more detail.
>
>
> Q2 (convergence) Thank you for noting this. We intended 'rapid convergence' to mean an increase in the derivative at zero, rather than a superior approximation of the unity function. We have revised the text to eliminate this ambiguity.
>
> The derivative at zero grows more rapidly under a composition of different polynomials due to its dependence on $\delta$. Specifically, the derivative of $q(d, \delta)$ increases with $\delta$. Since the sequence $\delta, \delta_1, \dots, \delta_l$ is monotonically increasing, the derivative also increases at each step. This ensures that the derivative of the compositional polynomial is strictly greater than that of a single, iterated polynomial.
>
> Q2 (algorithm) Thank you for noting this! There is a typo in algorithm of $\delta$-orthogonalization, it should be $X=p(X)$ in the penultimate line, and we can start with $A_r=1-\delta$. It will then take fewer iterations of binary search to reach the same result. We have corrected the algorithm in the revised text.
>
> Q3. It is a good question. The authors of Muon stated, that it is enough to make singular values fall into the interval $[0.7, 1.3]$, and more precise orthogonalization within their polynomial framework does not affect convergence in experiments with small NanoGPT. Our experiments, however, show, that polynomials with higher derivative at zero do have positive effect on the convergence. So in case of Muon, there should be some balance between proximity to 1 and the value of derivative.
>
> Concerning the Riemannian optimization on Stiefel manifold, we need exact orthogonalization for making retraction. Riemannian optimization has many applications in machine learning and beyond.

---

### Official Review · Reviewer_6zcm · 2025-10-30

**Soundness:** 4
**Presentation:** 3
**Contribution:** 3
**Rating:** 6
**Confidence:** 3

**Summary:**

Authors look into finding optimal polynomials for Newton-Schulz iterations that make matrices approximately orthogonal, which is a key step in the Muon optimizer for LLMs and is useful in optimization over the Stiefel manifold. They also propose to use an alternative cheap pre-normalization for the algorithm to work differing from the Frobenius norm. Experimental results show an improvement over previous approaches

**Strengths:**

Through rigorous theory, optimal polynomials are found for this task (sections 3 and 4)

A trick is proposed for using Gelfand's formula with almost no computational overhead in order to get accurate upper bounds on the spectral norm.

**Weaknesses:**

The theory seems to me to be more like a corollary of prior studies, but this does not necessarily undermine the value of this approach in this context.

Polynomials better fit to the task

See the questions/suggestions section for more.

typo in appendix K, it references Figure 1 instead of Figure 2

**Questions:**

Experiments with and without the Gelfand bound were performed, but I am missing info on directly what the Gelfand bound was versus the Frobenius norm approach to see directly what advantage it is providing for the initial estimate.

Also, it would have been good to isolate the effect of your different polynomials vs Gelfand's formula (you run Muon with the Frobenius norm bound and your approach is run using Gelfand's so both effects are mixed)

There could be more explanations on Figure 5. Several plots are show by number of steps despite that each line curve used a different number of matrix multiplications per step. The only comparable ones are Muon vs Cans order=5 iter=4, (this was mentioned in the text) and also Jiacheng iter 6 mm=18 vs CANS order 3 iter=9. This ploto should have been drawn with number of matrix multiplications in the x axis directly, not number of steps.

Also, a few more comparisons would be needed with Jiacheng work in order to see that this comparison was not hand picked

---

> ### Author Response · Authors · 2025-11-21
>
> Dear Reviewer  6zcm, thank you for taking time to review our article.
>
> Q1. The Frobenius norm considerably overestimates the largest singular value of the matrix. For example, in our experiment, in Figure 2, the Frobenius norm of the matrix is 999 and the largest singular value is 63. The table below shows that the number of matmuls either increases or does not change if we normalize the matrix by the Frobenius norm.
>
> \begin{array}{l|ll}
> Polynomials & Gelfand & Frobenius \\\\
> \hline
> Classical NS & 56 & 68\\\\
> CANS, degree 3, a_0=1e-3 & 36 & 48\\\\
> CANS, degree 5, a_0=1e-3 & 36 & 48\\\\
> CANS, degree 3, a_0=1e-7 & 44 & 44\\\\
> CANS, degree 5, a_0=1e-7 & 42 & 42 \\\\
> CANS, degree 3, \delta=0.99 & 30 & 42\\\\
> CANS, degree 5, \delta=0.99 & 33 & 36 \\\\
> \end{array}
>
> Q2. Thank you for this comment. We have added a comparison of different normalizations to Appendix L. The Figure 6 shows that Muon with Gelfand's normalization has slightly improved convergence compared to Frobenius normalization.
>
> Q3. We report the convergence versus the number of training steps in the Figure 5, because the number of steps is proportional to the wall-clock runtime. For Muon with all tested polynomials the training step takes approximately the same time, because the runtime overhead of orthogonalization in Muon is negligible.
>
> Theoretically this can be explained as follows. The FLOP overhead of Muon over SGD is $(T/3)m/B$ (see runtime analysis in [1]), where $m$ is matrix dimension, $B$ - sequence length, by $T$ we will denote number of matmuls ($T=15$ for original Muon). The difference in overhead of Muon with polynomials with $T_1$ and $T_2$ matmuls is $((T_1-T_2)/3)m/B$. In our experiment with NanoGPT, m=768, B=524288, the difference with original Muon is $T_1-T_2\leq 3$ so overhead is $((T_1-T_2)/3)m/B\leq 0.0015$.
>
> The table below presents time (in seconds, measured on A100 in bfloat16) required for orthogonalization of matrices of different sizes using polynomials with different number of matmuls. The time does not differ much depending on number of matmuls for sizes $\leq 2048$.
>
> \begin{array}{l|lllll}
> Matmuls & 256 & 512 & 1024 & 2048 & 4096 \\\\
> \hline
> 12 & 0.12 & 0.14 & 0.30 & 1.15 & 6.97  \\\\
> 14 & 0.15 & 0.18 & 0.38 & 1.40 & 8.35  \\\\
> 15 & 0.14 & 0.17 & 0.37 & 1.43 & 8.70  \\\\
> 18 & 0.17 & 0.21 & 0.44 & 1.71 & 10.48
> \end{array}
>
> Q4. The coefficients for this polynomial were algorithmically generated by the code from [2]. We fixed only the number of matrix multiplications, allowing the algorithm to yield the most accurate polynomial possible for our comparison and ensuring an objective selection. For completeness, we also add a polynomial with less matmuls (see the updated Figure 5).
>
> [1] Jordan, Keller, et al. "Muon: An optimizer for hidden layers in neural networks, 2024."
>
> [2] https://gist.github.com/YouJiacheng/393c90cbdc23b09d5688815ba382288b/5bff1f7781cf7d062a155eecd2f13075756482ae (see coefficients in https://docs.modula.systems/algorithms/newton-schulz/)

---

### Author Response · Authors · 2025-12-03

We are grateful to the reviewers for their thoughtful and constructive feedback. For the Area Chair’s convenience, the following summary outlines our contributions, primary responses and the corresponding revisions made to the paper.
## Contribution summary
Our primary contribution is a general framework for constructing Chebyshev-type polynomials that enable approximate orthogonalization under specified approximation and derivative constraints. We believe this approach provides a reusable foundation, representing a meaningful advance for Muon-style optimization algorithms and suggesting broader applicability in domains such as Riemannian optimization on the Stiefel manifold.
## Answer to reviewers
### 1. $\delta$-orthogonalization
Concerns.
* The words “rapid convergence” caused misunderstanding. Reviewer H3Bj (Q2) asked, what it means and why it’s better to use different polynomials. How should $\delta$ be chosen? (xmCb Q)

Our answer.
* These polynomials have two objectives: approximation of the unity function (parameter $\delta$) and the derivative at the origin, which is responsible for pushing small singular values to 1. By “rapid convergence” in that context, we initially meant increasing the derivative at zero, not tighter approximation to 1. We have rewritten the paragraph in section 4 about $\delta$ -orthogonalization to eliminate this ambiguity and add clarity. Composing polynomials with increasing $\delta_i$ yields a monotonically increasing derivative at 0, which accelerates the growth of small singular values. We have explained the algorithm for composing these polynomials and selecting $\delta$ in the same section.

### 2. Muon Experiments
Concerns.
* How to interpret the trade-off between closeness to 1 and derivative at 0? (6mHR Q6)
* Do we really need exact orthogonalization in for Muon? (H3Bj Q3)

Our answer.
* The authors of Muon observed that pushing singular values into $(1-\delta,1+\delta)$, while having large derivatives at 0 is a working strategy. So the key trade off is the value of $\delta$ vs derivative at 0, CANS polynomials  allow to improve it.
* Beyond Muon, accurate orthogonalization is necessary for Stiefel retractions, where our framework directly yields faster and scalable retractions.


### 3. Gelfand normalization
Concerns.
* How does the Gelfand estimate compares to Frobenius norm? Reviewer 6zcm asked to isolate the effect of normalization on convergence of Muon. (6zcm Q1–Q2)
* Reviewer H3Bj (Q1) asked about overhead of Gelfand’s formula (extra matmuls/stored matrices).

Our answer.
* Frobenius norm can severely overestimate the largest singular value, which either worsens or leaves unchanged the number of matmuls needed for orthogonalization. We added an experiment (Appendix L) showing that Gelfand normalization has positive effect on convergence of Muon.
* We added explanation about Gelfand’s formula implementation to the Section 3.3, for degree 3 there is no overhead at all, while for degrees >3, we can save intermediate powers, thus introducing no extra matmuls. Note that normalization is applied only on the first step, which in any case introduces negligible overhead relative to the whole orthogonalization process.

### 4. Stiefel Manifold Experiments and Generality
Concerns.
* Experiments only on Wide-ResNet 16 10 / CIFAR 10; how general is CANS retraction?

Our answer.
* This setup aligns with prior Cayley retraction benchmark. We stress that CANS is a general retraction and added new timing experiments (Appendix M) showing that CANS scales better than Cayley and QR retractions for large thin matrices (common in LoRA), making it a broadly applicable, more efficient primitive.

---

### Meta-Review · Area_Chair_kmyd · 2026-01-19

**Summary:**

This paper proposes Chebyshev-Accelerated Newton-Schulz, a framework for optimizing Newton-Schulz iterations using Chebyshev polynomials to compute approximate matrix orthogonalizations more efficiently.

- Reviewer 6zcm raised concerns that the experimental results mix the effects of polynomial optimization and Gelfand pre-normalization.

- Reviewer H3Bj raised concerns regarding the limited discussions of concurrent work.

- Reviewer xmCb's concerns are about the limited scope of experiments and insufficient guidance for selecting the $\delta$ parameter,

- Reviewer 6mHR's concerns include the overly technical presentation, lack of high-level intuition, and incomplete reporting of Muon optimizer results.

**Reviewer Concerns:**

The authors have mainly addressed the concerns from Reviewers 6zcm, H3Bj, and xmCb. However, reviewer 6mHR is not convinced by the experiments comparing Muon and other optimizers, noting that the theoretical advantage in optimization is not clearly reflected in the performance curves.

**Reviewer Scores:**

Even with full discussion, reviewer scores would likely remain largely unchanged, as concerns regarding experimental clarity, limited scope, and unclear practical impact would persist, supporting a rejection decision.

---

### Decision · Program_Chairs · 2026-01-26

Reject